# `MetaDistill`: Unlocking the Performance Ceiling for Pretrained Optimizers

Muqi Han [* 1 2]  Ruoqi Xing [* 3 4]  Kai Wu [5]  Xiaoyu Zhang [6]  Handing Wang [5]  Zilong Wang [6]

## Abstract

Meta Black-Box Optimization (MetaBBO) has emerged as a promising paradigm by employing meta learning to automatically optimize the configurations of low-level black-box optimizers. Despite its potential, the generalization of MetaBBO remains significantly constrained when facing unseen, complex objective landscapes. We identify that this bottleneck stems from a restricted performance upper bound inherent in current training mechanisms: by learning from scratch in a self-supervised or unsupervised manner, low-level learnable optimizers are never exposed to advanced, high-quality optimization behaviors, forcing them to converge on suboptimal strategies. In this paper, we propose `MetaDistill`, a general MetaBBO training framework designed to lift the strategy ceiling through pretraining and test-time fine-tuning. In the pretraining stage, we represent high-quality strategies from classical algorithms as expert optimization trajectories and utilize them for diversity-preserving distillation, enabling the learnable optimizer to internalize advanced optimization behaviors. In the optional fine-tuning stage, we perform self-supervised fine-tuning as a warm-start procedure to further refine the distilled knowledge on unseen tasks. We evaluate our `MetaDistill` framework on the BBOB test suite and three control tasks. The results demonstrate that `MetaDistill` significantly improves the generalization ability of various learnable optimizers compared to their original training paradigms. Our code is available at here.

---

[*]Equal contribution [1]Guangzhou Institute of Technology, Xidian University, China [2]The State Key Laboratory of Blockchain and Data Security, Zhejiang University, China [3]School of Mathematics and Statistics, Xidian University, China [4]Key Laboratory of Data and Intelligent System Security, Ministry of Education, China [5]School of Artifcial Intelligence, Xidian University, China [6]School of Cyber Engineering, Xidian University, China. Correspondence to: Kai Wu <kwu@xidian.edu.cn>.

*Proceedings of the 43rd International Conference on Machine Learning*, Seoul, South Korea. PMLR 306, 2026. Copyright 2026 by the author(s).

## 1. Introduction

Adaptive black-box optimization methods, such as CMA-ES (Hansen, 2016), JADE (Zhang & Sanderson, 2009), and L-SHADE (Tanabe & Fukunaga, 2014), outperform their non-adaptive counterparts, such as ES (Hansen et al., 2015) and DE (Storn & Price, 1997), on many complex problems. However, the performance gains come at the cost of not only the need for meticulous handcrafted optimization operators, but also an increased number of parameters. This makes it difficult to transfer parameter configurations adapted to one task to new tasks, thus limiting their cross-task optimization capabilities.

To address this issue, Meta Black-Box Optimization (MetaBBO) (Ma et al., 2025b) has emerged as a promising paradigm. By leveraging meta learning, MetaBBO seeks to automatically internalize optimization rules from a distribution of pretraining tasks, enabling rapid adaptation to unseen problems with minimal human intervention. Current MetaBBO research primarily follows two trajectories. The first and most prevalent category employs self-supervised or unsupervised objectives to discover optimization rules from scratch. For instance, POM (Li et al., 2024) utilizes fitness improvements between successive generations to update parameters via gradient descent, while LES (Lange et al., 2023b) and Q-Mamba (Ma et al., 2025a) use evolutionary strategies or reinforcement learning to search for optimal meta-level configurations.

Despite their promise, we identify a fundamental flaw in this scratch-based paradigm: the existence of a restricted performance upper bound. When meta-optimizers learn through pure trial-and-error without external guidance, they are never exposed to advanced, high-quality optimization behaviors. In complex landscapes, this lack of expertise causes the learning process to be plagued by local optima and poor initializations, forcing the model to settle for suboptimal strategies. Essentially, while these models learn how to optimize, they fail to capture the deep optimization wisdom required to navigate the most challenging objective frontiers.

A second category of MetaBBO attempts to bridge this gap through supervised learning or knowledge distillation (Hinton et al., 2015). However, existing efforts remain limited in scope and generality. Evolutionary Algorithm

Distillation (EAD) (Lange et al., 2024) introduces a "single-teacher, single-student" framework but restricts itself to Gaussian-based strategies, thereby failing to capture the rich, heterogeneous strategies present in the broader BBO ecosystem. This limitation suggests that the strategy ceiling cannot be shattered by simply imitating a single algorithm; it requires a systematic way to ingest diverse, multi-expert expertise.

To bridge this gap and lift the strategy ceiling, we propose `MetaDistill`, a novel training framework that shifts the MetaBBO paradigm from learning through trial-and-error to learning from expert roadmaps. We recognize that classical BBO algorithms possess decades of embedded wisdom; by representing these strategies as high-quality expert trajectories, we provide a roadmap for the meta-learner to internalize advanced behaviors. `MetaDistill` introduces a diversity-preserving distillation (DPD) mechanism that infuses the learnable optimizer with multi-expert knowledge during pretraining, effectively elevating its optimization upper bound beyond what any scratch-based method can achieve. Extensive evaluations on the BBOB test suite demonstrate that `MetaDistill` consistently shatters the performance ceiling of existing learnable optimizers, yielding significant gains in both convergence speed and cross-task generalization.

Our primary contributions are summarized as follows:

- **A Paradigm Shift in Training**: We introduce the `MetaDistill` framework, challenging the *scratch-based* dominance by treating classical optimization wisdom as a supervised roadmap for MetaBBO.

- **Flexible Distillation & Adaptation**: We design a framework that supports multi-teacher, any-student distillation and is compatible with varying population sizes. We further integrate a self-supervised fine-tuning (SSFT) stage to refine distilled knowledge for specific unseen nuances.

- **Counter-intuitive Empirical Insights**: We uncover a *diversity paradox* through extensive parametric studies, revealing that excessive task or teacher diversity can hinder generalization. By identifying architecture-specific saturation points, we provide a practical blueprint for data-efficient MetaBBO training.

## 2. Related Work

### 2.1. Unsupervised and Self-Supervised Meta Black-Box Optimization

MetaBBO (Ma et al., 2025b) aims to meta-train a learnable optimizer over a distribution of tasks, enabling it to autonomously acquire optimization strategies and rapidly adapt to unseen but related problems. Existing MetaBBO approaches predominantly rely on unsupervised or self-supervised objectives to train low-level optimizers. For instance, Meta-MOGA (Li et al., 2025a), LES (Lange et al., 2023b) and LGA (Lange et al., 2023a) utilize a meta-level black-box optimizer to evolve low-level optimizers, using metrics like hypervolume or aggregated fitness as meta-objectives. Conversely, B2Opt (Li et al., 2025b), POM (Li et al., 2024), EPOM (Han et al., 2026), and ABOM (Wang et al., 2026) design self-supervised objectives based on fitness improvements between consecutive generations, allowing the optimizer to learn evolutionary dynamics in a self-boosting manner. From a reinforcement learning (RL) perspective, LDE (Sun et al., 2021), ConfigX (Guo et al., 2025) and Q-Mamba (Ma et al., 2025a) formulate MetaBBO by constructing self-supervised rewards, typically derived from observed performance gains during the optimization trajectory.

### 2.2. Supervised Meta Black-Box Optimization

In contrast, several recent studies incorporate explicit supervision, with many leveraging autoregressive models to learn solution trajectories. For example, BONET (Mashkaria et al., 2023) and RIBBO (Song et al., 2024) employ Transformers (Vaswani et al., 2017) to generate candidate solutions on refined offline datasets without querying the objective function during the generation phase. OptFormer (Chen et al., 2022) serves as a hyperparameter optimizer by jointly imitating multiple search policies and predicting objective function distributions across diverse datasets. While these methods all involve learning from collected trajectories, their objectives differ significantly: BONET and RIBBO focus on offline BBO; and OptFormer targets hyperparameter optimization. In contrast, our proposed method focuses on elevating the strategy ceiling and enhancing the generalization capabilities of population-based learnable optimizers.

### 2.3. Learn to Optimize

MetaBBO and L2O both study meta-trained learnable optimizers (LOs) for black-box and gradient-based optimization, respectively. Existing L2O research covers gradient-based neural update rules trained with Evolution Strategies (Metz et al., 2022b), stable width scaling via $\mu$-parameterization (Thérien et al., 2026), performance–efficiency trade-offs (Metz et al., 2022a), training stability through inductive biases and hybrid gradient estimation (Harrison et al., 2022; Metz et al., 2019), cross-task generalization with flatness-aware meta-objectives (Yang et al., 2023), and communication-efficient distributed learning with meta-learned optimizers (Joseph et al., 2025).

## 2.4. Algorithm Distillation

Algorithm Distillation (AD) (Laskin et al., 2023) was originally proposed to train neural networks on the learning histories of RL algorithms via knowledge distillation (Hinton et al., 2015). Evolutionary Algorithm Distillation (EAD) (Lange et al., 2024) extends this paradigm to evolutionary computation by training Evolution Transformers to mimic Gaussian-based evolutionary strategies. Although both EAD and `MetaDistill` employ distillation to train learnable optimizers, EAD is confined to specific Gaussian-based strategies. Our `MetaDistill`, however, introduces a strategy-agnostic trajectory distillation framework, offering greater flexibility and applicability across diverse optimization landscapes.

## 3. Methodology

### 3.1. Problem Definition

Black-box optimization (BBO) involves seeking an optimal solution $\mathbf{x}^\star$ that minimizes an objective problem $f$, where the internal properties of $f$, such as derivatives and landscape geometry are typically unavailable:

$$\mathbf{x}^\star = \arg\min_{\mathbf{x}\in\mathbb{R}^d} f(\mathbf{x}) \qquad (1)$$

where $d$ denotes the dimensionality of the search space. Under this formulation, any operational constraints can be optionally incorporated into $f$ via penalty functions, thereby recasting the problem as an unconstrained optimization task.

meta black-box optimization extends this paradigm by aiming to learn a high-level configuration or policy for a low-level optimizer. Specifically, it seeks to train a learnable optimizer $\mathcal{A}(\theta)$ parameterized by $\theta$, across a distribution of tasks $\mathcal{F}$. The objective is to identify the optimal parameters $\theta^\star$ that yield the best expected performance over $\mathcal{F}$, facilitating rapid adaptation to unseen but related downstream tasks:

$$\theta^\star = \arg\min_{\theta\in\Theta} \mathbb{E}_{f\sim\mathcal{F}} \left[ \mathcal{L}\left( \mathcal{A}(\theta; f)\right)\right] \qquad (2)$$

where $\mathcal{L}(\cdot)$ represents a performance metric achieved by $\mathcal{A}$ on task $f$.

In conventional MetaBBO, $\mathcal{L}$ is typically formulated as a self-supervised or unsupervised objective, such as the fitness difference between adjacent generations or the the final objective value. However, we contend that such objectives often restrict the performance ceiling of $\mathcal{A}$. Particularly in the nascent stages of training, deficient optimization capabilities can trap the learnable optimizer in local optima or lead to premature convergence, ultimately compromising generalization on unseen tasks. To circumvent these limitations, we introduce `MetaDistill`, a framework that leverages external optimization knowledge to construct a strongly supervised objective, $\mathcal{L}_1$. This supervision facilitates a rapid

enhancement of the optimizer's foundational capabilities. Building upon this, we further incorporate a self-supervised objective, $\mathcal{L}_2$, tailored for downstream tasks to fine-tune the optimizer and bolster its generalization performance.

---

**Algorithm 1** Diversity-Preserving Distillation
---

**Input:** Student learnable optimizer $\mathcal{A}(\theta)$ parameterized by $\theta$ and teacher trajectory set $TS = \left[ [\tau_{m,f_n}]_{n=1}^{|\mathcal{F}|} \right]_{m=1}^{|TS|}$.

**Output:** The optimal $\theta^\star$.

  **while** not convergence **do**

    Sample $[\tau_{m,f_n}]_{n=1}^{|\mathcal{F}|}$ from $TS$

    **for** $t = 1, 2, \ldots, T$ **do**

      **for** $n = 1, 2, \ldots |\mathcal{F}|$ **do**

        $\mathbf{X}^t \leftarrow \mathcal{A}(\mathbf{X}^{t-1}, f_n|\theta)$

        Calculate $\mathcal{L}_1$ using Eq.(4)

        Update $\theta$ by $\nabla_\theta \mathcal{L}_1$

      **end for**

    **end for**

  **end while**

---

### 3.2. `MetaDistill`

We introduce the `MetaDistill` framework to optimize a learnable optimizer $\mathcal{A}(\theta)$ by distilling knowledge from established optimization heuristics.

**Trajectories Collection** As illustrated in Fig. (1), the process begins by harvesting offline data from multiple classic BBO algorithms including DE (Storn & Price, 1997), JADE (Zhang & Sanderson, 2009), SHADE (Tanabe & Fukunaga, 2013), L-SHADE (Tanabe & Fukunaga, 2014), PSO (Kennedy & Eberhart, 1995), GA (Holland, 1992), and CMA-ES (Hansen, 2016). To ensure the quality of the distillation source, for these algorithms, we perform a fitness-based filtering, capturing the most effective optimization steps at intervals of $i$ generations to assemble a refined optimization trajectory. This procedure results in a composite trajectory of length $T$ used for distillation training, where $T$ represents the maximum generation.

**Distributional Alignment and Diversity-Preserving Distillation** We contend that the optimization knowledge of a BBO algorithm is implicitly embedded within its search strategy. Consequently, a natural approach to constructing a trajectory of optimization knowledge would be to archive these evolving strategies directly. However, within the "multi-teacher, any-student" paradigm, which aims to support architecture-agnostic distillation, this direct storage is impractical. The fundamental challenge lies in the structural heterogeneity of the teachers. For instance, if CMA-ES and JADE serve as teachers, the resulting trajectory $\tau = \left[ \ldots, \left( \mathbf{F}^i, \mathbf{CR}^i \right), \ldots, \left( \boldsymbol{\mu}^j, \boldsymbol{\Sigma}^j \right), \ldots \right]$, forces a paradigm-specific dependency. An ES-based student (e.g.

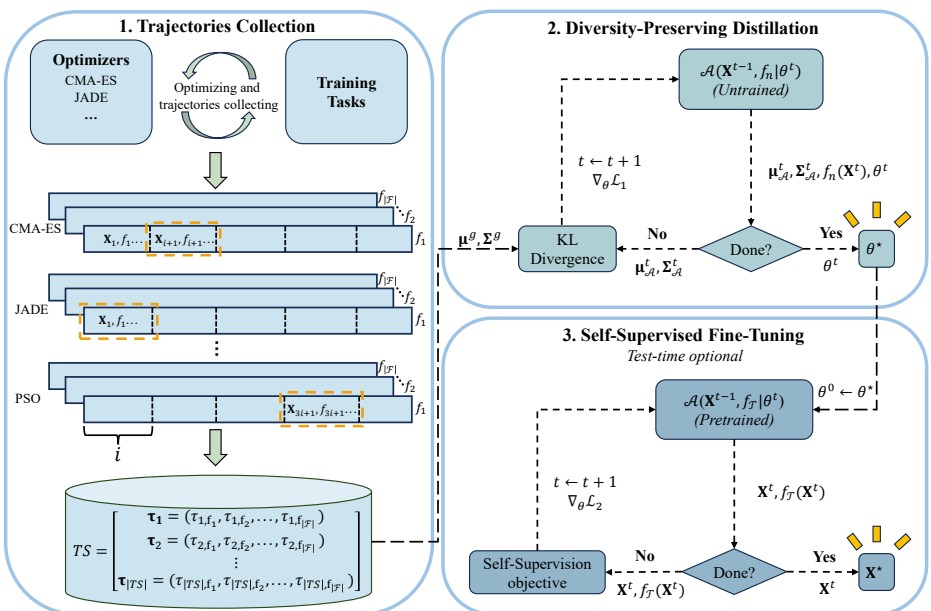

*Figure 1.* Overview of the `MetaDistill` framework. (Left) The pipeline begins with a collection of optimization trajectories $TS$ by executing various black-box optimizers on a task set $\mathcal{F}$. (Top right) A learnable optimizer $\mathcal{A}(\theta)$ is then trained via diversity-preserving distillation to effectively capture these trajectories. (Bottom right) Finally, an optional SSFT strategy can be employed during inference to augment the adaptability of the pretrained optimizer $\mathcal{A}(\theta^\star)$ to unseen downstream tasks.

LES Lange et al. 2023b) is inherently limited to mimicking the Gaussian parameters of CMA-ES, while a DE-based student (e.g. POM Li et al. 2024) can only process the **F** and **CR** updates from JADE. Such a representation mismatch prevents a universal student from effectively aggregating knowledge from diverse optimization experts.

To address this challenge, we leverage the population—a universal medium common to all Evolutionary Algorithms and Swarm Intelligence algorithms—to facilitate architecture-agnostic distillation. Instead of archiving algorithm-specific parameters, we represent the optimization process through population trajectories. Building upon the fitness-based filtering described previously, we represent each generation within the selected trajectory segments by its mean and covariance matrix. This transformation projects the heterogeneous optimization strategies of diverse teachers into a unified distribution space, yielding a standardized training trajectory: $\tau = \left[\left(\boldsymbol{\mu}^1, \boldsymbol{\Sigma}^1\right), \left(\boldsymbol{\mu}^2, \boldsymbol{\Sigma}^2\right), \ldots, \left(\boldsymbol{\mu}^T, \boldsymbol{\Sigma}^T\right)\right]$. The distillation process employs the forward Kullback-Leibler (KL) divergence as the loss function. This allows the student to capture the generative patterns of diverse teachers on problem $f$ without being constrained by their specific architectural

implementations:

$$\mathbf{X}^t = \mathcal{A}\left(\mathbf{X}^{t-1}, f|\theta\right)$$

$$\boldsymbol{\mu}_{\mathcal{A}}^t = \frac{1}{N}\sum_{i=1}^{N}\mathbf{x}_i^t$$

$$\boldsymbol{\Sigma}_{\mathcal{A}}^t = \frac{1}{N-1}\sum_{i=1}^{N}\left(\mathbf{x}_i^t - \boldsymbol{\mu}_{\mathcal{A}}^t\right)^{\top}\left(\mathbf{x}_i^t - \boldsymbol{\mu}_{\mathcal{A}}^t\right)$$

$$\boldsymbol{\mu}^g, \boldsymbol{\Sigma}^g = \tau^g \tag{3}$$

$$\hat{\mathcal{L}}_1 = \frac{1}{2}\left[\log\frac{|\boldsymbol{\Sigma}^g|}{|\boldsymbol{\Sigma}_{\mathcal{A}}^t|} - d + \mathrm{tr}\left(\left(\boldsymbol{\Sigma}^g\right)^{-1}\boldsymbol{\Sigma}_{\mathcal{A}}^t\right)\right]$$
$$+ \frac{1}{2}\left[\left(\boldsymbol{\mu}_{\mathcal{A}}^t - \boldsymbol{\mu}^g\right)\left(\boldsymbol{\Sigma}^g\right)^{-1}\left(\boldsymbol{\mu}_{\mathcal{A}}^t - \boldsymbol{\mu}^g\right)^{\top}\right],$$

Here, $g = g(t; i, T) = \min\left(\lceil\frac{t}{i}\rceil \times i, T\right)$ denotes the target generation for distillation, which effectively creates a temporal window of length $i$ that governs the distillation schedule and slides forward every $i$ generations. This formulation ensures that for any generation $t$ within the same window, the student distribution $(\boldsymbol{\mu}_{\mathcal{A}}^t, \boldsymbol{\Sigma}_{\mathcal{A}}^t)$ consistently targets the reference generation $\tau^g$. Such curriculum-based distillation enables the learnable optimizer to emulate the teachers' strategies in a more stable and progressive manner.

However, in practice, directly minimizing the KL divergence $\hat{\mathcal{L}}_1$ as defined in Eq. (3) often triggers numerical instability,

particularly when the teacher and student distributions exhibit significant discrepancy. To address this, we introduce a dual-stage scaling scheme using coefficients $s_1$ and $\mathbf{s}_2$. Specifically, $s_1$ is the reciprocal of the square root of the dimension $d$. To further constrain the scale of the gradients, we define $\mathbf{s}_2$ based on the element-wise maximum absolute value of the respective mean vectors and covariance matrices:

$$
\begin{aligned}
s_1 &= \frac{1}{\sqrt{d}} \\
s_2^{\mu} &= \frac{1}{\max\left(\max\left(\mathrm{abs}\left(\boldsymbol{\mu}_{\mathcal{A}}^{t}\right)\right), \max\left(\mathrm{abs}\left(\boldsymbol{\mu}^{g}\right)\right)\right)} \\
s_2^{\Sigma} &= \frac{1}{\max\left(\max\left(\mathrm{abs}\left(\boldsymbol{\Sigma}_{\mathcal{A}}^{t}\right)\right), \max\left(\mathrm{abs}\left(\boldsymbol{\Sigma}^{g}\right)\right)\right)} \\
\mathcal{L}_1 &= s_1 \cdot \hat{\mathcal{L}}_1\left(s_2^{\mu} \cdot \boldsymbol{\mu}_{\mathcal{A}}^{t}, s_2^{\Sigma} \cdot \boldsymbol{\Sigma}_{\mathcal{A}}^{t} \| s_2^{\mu} \cdot \boldsymbol{\mu}^{g}, s_2^{\Sigma} \cdot \boldsymbol{\Sigma}^{g}\right).
\end{aligned}
\tag{4}
$$

This distribution-centric approach offers three distinct advantages: *1) Population Size Agnostic Training*: Distribution-level matching enables `MetaDistill` to bridge teachers and students with different population sizes seamlessly. *2) Generalization over Rigid Imitation*: By distilling the underlying distribution, the student avoids point-to-point overfitting to specific teacher trajectories, promoting the acquisition of broader optimization principles. *3) Diversity Preservation*: Due to the mass-covering property of the forward KL divergence, the student is not penalized for exploring regions where the teacher's density is zero. This encourages the student to maintain high population diversity rather than collapsing into a narrow mode. The pseudo-code for DPD is shown in Algorithm (1).

---

**Algorithm 2** Self-Supervised Fine-Tuning

---

**Input:** Learnable optimizer $\mathcal{A}(\theta)$ parameterized by $\theta$, $\mathcal{F}$-optimal $\theta^{\star}$, unseen task $f_{\mathcal{T}}$.
**Output:** Optimal $\mathbf{X}^{\star}$ for $f_{\mathcal{T}}$.
 $\theta \leftarrow \theta^{\star}$
 **for** $t = 1, 2, \ldots, T$ **do**
  $\mathbf{X}^t \leftarrow \mathcal{A}(\mathbf{X}^{t-1}, f_{\mathcal{T}} | \theta)$
  Calculate $\mathcal{L}_2$ using Eq.(5) or Eq.(6)
  Update $\theta$ by $\nabla_\theta \mathcal{L}_2$
 **end for**

---

**Self-Supervised Fine-Tuning** Complementing DPD, we propose an optional test-time self-supervised fine-tuning strategy to enhance the adaptability of the pretrained optimizer $\mathcal{A}(\theta^{\star})$ to downstream tasks by utilizing the optimization heuristics and inherent task properties encountered during the inference process.

*Differentiable Objective.* For a differentiable problem $f^D$, inspired by MetaGBT (Li et al., 2024), we construct a self-supervised objective $\mathcal{L}_2^D$ based on the relative improvement

in fitness. This objective monitors the mean fitness discrepancy between the current population $\mathbf{X}^t$ and a historical population $\mathbf{X}^{t-j}$, thereby generating a gradient signal that steers the search toward better fitness values:

$$
\mathcal{L}_2^D = \frac{\mathbb{E}\left[f(\mathbf{X}^t) - f(\mathbf{X}^{t-j})\right]}{|\mathbb{E}\left[f(\mathbf{X}^{t-j})\right]|},
\tag{5}
$$

where $j = 1, 2, \ldots, t-1$ is self-supervision interval. For the sake of notational brevity, we adopt $\mathbb{E}[\cdot]$ as a shorthand for the empirical mean of a vector rather than using the summation operator $\sum$.

*Black-Box Objective.* However, in practice, most downstream tasks are black-box problems ($f^B$), where the unavailability of $\nabla f$ renders the fitness derivative based objective $\mathcal{L}_2^D$ inapplicable. To circumvent this, we once again leverage the population, whose differentiability depends strictly on the optimization strategy rather than the objective function. Specifically, we design a population-wise self-supervised objective that aligns the current search distribution with the observed elite individuals:

$$
\mathcal{L}_2^B = \mathbb{E}\left[\mathrm{abs}\left(\frac{\boldsymbol{\mu}_{\mathbf{C}}^t - \boldsymbol{\mu}_{\mathbf{E}}^t}{\boldsymbol{\mu}_{\mathbf{E}}^t}\right)\right],
\tag{6}
$$

where $\boldsymbol{\mu}_{\mathbf{C}}^t$ denotes the empirical mean of the candidate population (e.g., prior to selection) at generation $t$, while $\boldsymbol{\mu}_{\mathbf{E}}^t$ represents that of the elite population (e.g., post-selection) at the same generation. As indicated by Eq (6), this objective remains consistently valid for black-box tasks provided the underlying optimization strategy is differentiable. By minimizing this discrepancy, the pretrained optimizer is encouraged to shift its generative distribution toward the low-fitness regions identified by the selection operator. The pseudo-code for self-supervised fine-tuning is shown in Algorithm (2). For computational efficiency, we introduce the self-supervision interval $j$, which governs the frequency of parameter updates for the pretrained optimizer. Analogous to its role in differentiable objective, $j$ here defines the adaptation granularity. Since black-box and differentiable objectives are mutually exclusive, we employ this unified notation to simplify the transition between the two contexts.

## 4. Experiments

### 4.1. Experimental Setup

We evaluate the performance of four representative learnable optimizers: LES (Lange et al., 2023b), LGA (Lange et al., 2023a), LDE (Sun et al., 2021), and POM (Li et al., 2024)—trained via our proposed `MetaDistill` framework against their respective original training paradigms (denote as baselines). Evaluations are conducted on the BBOB test suite (Hansen et al., 2021) and three control tasks.

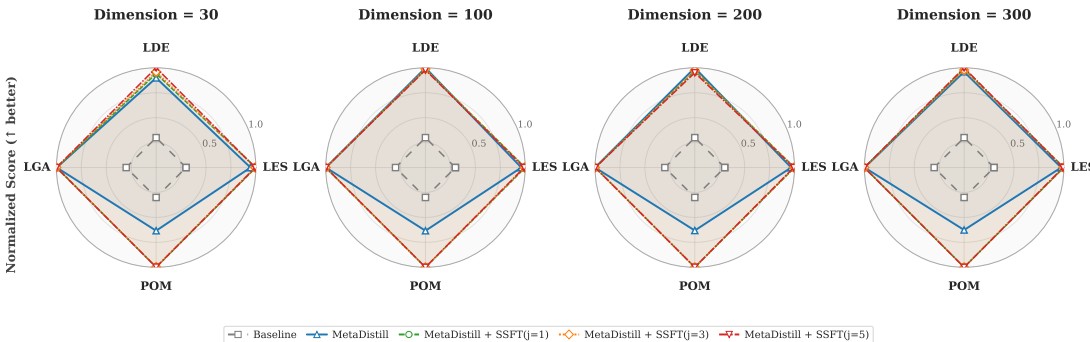

*Figure 2.* Normalized relative performance differences between the original versions (Baselines) and `MetaDistill` variants of four representative learnable optimizers on BBOB with dimension settings of 30, 100, 200 and 300.

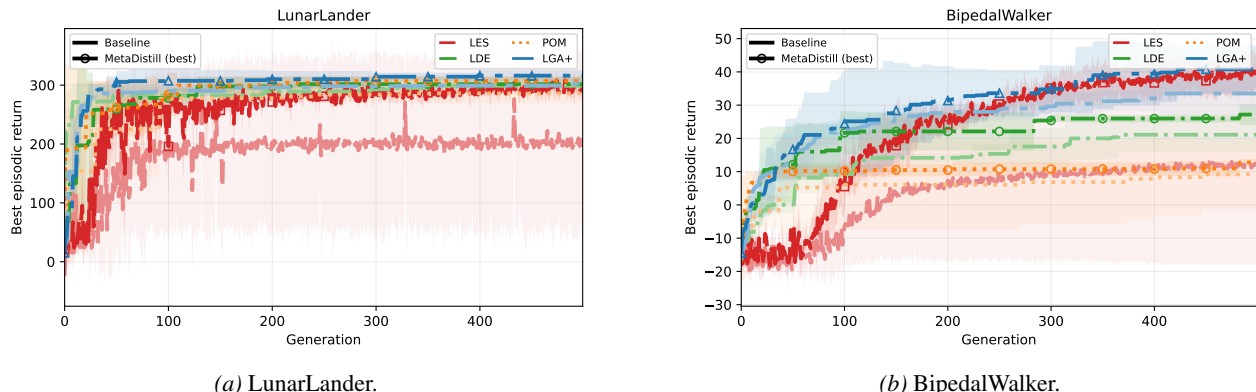

| *(a)* LunarLander. | *(b)* BipedalWalker. |

*Figure 3.* Best episodic return versus generation. Lines show mean over 3 seeds; shaded areas indicate $\pm 1$ std. Solid: baseline. Dashed: best `MetaDistill` variant. Higher is better.

During training, we use five synthetic functions (TF1 to TF5 in Table 6) as training tasks, the dimension for all functions is set to $d = 10$, the population size is set to $N = 200$, and the epoch length is set to $H = 128$, with an evaluation budget of 20,000 per function per epoch. For `MetaDistill`, we construct an additional knowledge base $TS$ comprising 80 optimization trajectories. These trajectories serve as the distillation source and are generated by running seven classic population-based BBO algorithms on the training tasks.

For comparison, the *baselines* of these optimizers refer to those trained using their native frameworks: MetaBBO (Lange et al., 2023b)[1] for LES and LGA, policy gradient (Sun et al., 2021) for LDE, and MetaGBT (Li et al., 2024) for POM. More details (i.e. optimizer configurations, expressions for all training tasks) are provided in Appendix A.

---
[1]Here, MetaBBO specifically refers to the training framework based on bi-level black-box optimization, rather than the broader meta black-box optimization community.

*Table 1.* Logarithmic Absolute Difference (LAD, see Appendix C for details) of the best `MetaDistill` variant over the baseline on BBOB $f_1-f_{24}$ (higher is better). **The best variant per cell is posteriorly selected from {md_j0, md_j1, md_j3, md_j5}.**

| $\mathcal{A}$ | LAD $\uparrow$ | | | |
| --- | --- | --- | --- | --- |
| | $d{=}30$ | $d{=}100$ | $d{=}200$ | $d{=}300$ |
| LES | 0.97 | 0.87 | 0.63 | 0.45 |
| LDE | 0.64 | 0.52 | 0.52 | 0.46 |
| LGA | 2.18 | 2.31 | 2.45 | 2.51 |
| POM | 5.41 | 5.47 | 5.46 | 5.41 |

### 4.2. Results

**BBOB** We evaluate the generalization performance of four baseline learnable optimizers against their `MetaDistill` variants across 24 BBOB functions, with dimensions set at $d \in \{30, 100, 200, 300\}$. Relative performance differences are illustrated in Figure 2, while quantitative data and supplementary results are provided in Table 1 and Appendix B, respectively. Based on these results, our

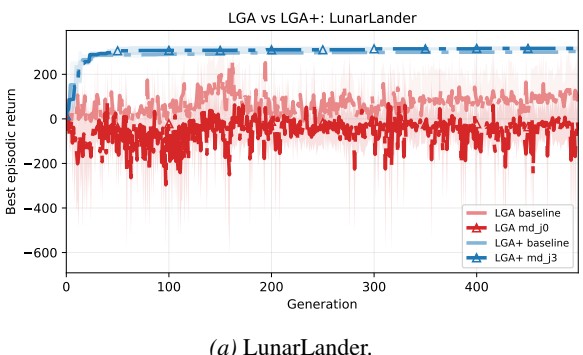 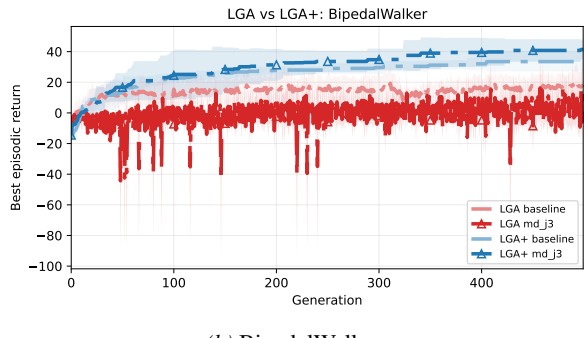

*(a)* LunarLander.    *(b)* BipedalWalker.

*Figure 4.* LGA vs. LGA+ on two control tasks. The original LGA suffers from premature convergence, while LGA+ with deterministic selection achieves strong performance. Solid: baseline; Dashed: best `MetaDistill` variant.

key findings are as follows:

- **POM** benefits most dramatically, achieving an $\sim 10^5\times$ reduction in the aggregated final objective value ($\log_{10}$ gain $\approx 5.4$) consistently across all dimensions.

- **LGA** shows strong improvement ($\sim 150$–$300\times$), with gains increasing at higher dimensions (from 2.18 at $d{=}30$ to 2.51 at $d{=}300$).

- **LES** and **LDE** exhibit more modest gains ($\sim 3$–$10\times$), with improvements decreasing slightly at higher dimensions.

- The optimal **self-supervision interval** $j$ varies by architecture and dimension. For example, LGA's best variant is `md_j3` at $d{=}30$, `md_j1` at $d{=}100$, `md_j0` at $d{=}200$, and `md_j5` at $d{=}300$; LES consistently prefers $j{=}1$; LDE often performs best with `md_j0` [2].

These results demonstrate that `MetaDistill` consistently elevates the performance ceiling of all four learnable optimizers. Furthermore, the integration of SSFT provides additional gains, enabling these models to not only surpass their baselines in most tests but also achieve performance on par with or exceeding the teacher's filtered trajectories.

**Control Tasks** We evaluate `MetaDistill` on three Gymnasium (Towers et al., 2026) control tasks by evolving an MLP of $d$ parameters as the control policy. In the main text, we highlight two representative tasks: **LunarLander** and **BipedalWalker**. LunarLander has an 8D observation and 4 discrete actions ($d = 420$), while BipedalWalker has a 24D observation and 4 continuous actions ($d = 932$). Complete experimental setup details, full results for all three tasks, and additional learning curves are provided in Appendix D.

Table 2 reports final episodic **return** for each optimizer, comparing its original version (baseline) to the best-performing `MetaDistill` variant on each task. For each (optimizer, task) pair, the best `MetaDistill` variant is selected from {`md_j0`, `md_j1`, `md_j3`, `md_j5`} by the highest mean return (see Appendix D.2 for the full grid).

Across both representative tasks, the optimal `MetaDistill` variants consistently enhance final returns, with the most pronounced improvements observed in configurations where the baselines are relatively weak. In BipedalWalker, `MetaDistill` significantly bolsters fragile baselines; for instance, LES not only boosts its return from 12.5 to 39.5 but also exhibits reduced variance. Similar gains are observed for POM, while LDE shows modest but steady improvement. In LunarLander, `MetaDistill` markedly upgrades LES and POM, though the improvement is less dramatic for the already-robust LDE baseline.

However, we observe that LGA suffers from premature convergence and even population degradation in both tasks (see Appendix D.3 for details). We attribute this failure to its learnable selection operator, which fails to capture a robust selection strategy in these environments. To test this hypothesis, we replace the learnable selection operator in both the baseline LGA and its `MetaDistill` variants with a deterministic $(\mu + \lambda)$ selection mechanism, ensuring that the elite individuals are preserved across generations.

As shown in Table 2 and Figure 3, this refined version (denoted LGA$^+$) achieves a significant performance leap, even surpassing the baselines and variants of all other learnable optimizers in LunarLander. Figure 4 further illustrates this contrast: while the original LGA collapses early in training, LGA$^+$ exhibits stable convergence and substantial `MetaDistill` gains. Based on these findings, we offer a critical insight for the design of population-based learnable optimizers: *The selection operator should remain concise*

---

[2]Here, `md_j`{z} refers to `MetaDistill` with SSFT interval $j = z$, and $z = 0$ means we do not perform SSFT.

*Table 2.* Final episodic return on the two focus tasks. We compare each optimizer's baseline to the best `MetaDistill` variant (posteriorly selected by highest mean return among `md_j{0,1,3,5}`).

| Optimizer | LunarLander | | BipedalWalker | |
| | Baseline | `MetaDistill` (best variant) | Baseline | `MetaDistill` (best variant) |
|---|---|---|---|---|
| LES | $195.627 \pm 152.976$ | $\mathbf{293.821 \pm 26.896}$ | $12.540 \pm 30.805$ | $\mathbf{39.469 \pm 6.377}$ |
| LDE | $298.954 \pm 7.088$ | $\mathbf{301.686 \pm 12.237}$ | $21.098 \pm 5.113$ | $\mathbf{27.154 \pm 3.119}$ |
| POM | $286.003 \pm 16.871$ | $\mathbf{307.141 \pm 7.822}$ | $9.171 \pm 10.524$ | $\mathbf{12.771 \pm 1.012}$ |
| LGA | $111.915 \pm 161.801$ | $-48.767 \pm 82.077$ | $18.401 \pm 12.714$ | $14.594 \pm 16.948$ |
| LGA$^+$ | $302.710 \pm 5.141$ | $\mathbf{316.111 \pm 9.534}$ | $33.495 \pm 6.462$ | $\mathbf{41.406 \pm 6.406}$ |

*Table 3.* Parameter study on BBOB at $d{=}100$, LAD is reported per cell. Bold indicates best per algorithm (within each block).

| $\mathcal{A}$ | Varying #tasks (`tsetK7`) | | | | | Varying #teachers (`fsetK16`) | | |
| | K=**1** | **3** | **5** | **10** | **16** | K=**1** | **3** | **7** |
|---|---|---|---|---|---|---|---|---|
| LGA | 1.786±0.007 | 1.800±0.004 | 1.808±0.003 | 1.807±0.003 | **1.813±0.003** | 1.798±0.002 | 1.799±0.003 | **1.813±0.003** |
| LDE | 0.191±0.013 | 0.333±0.018 | **0.373±0.021** | 0.339±0.010 | 0.347±0.017 | **0.358±0.027** | 0.331±0.022 | 0.347±0.017 |
| LES | 0.068±0.158 | 0.012±0.312 | **0.149±0.349** | -0.072±0.187 | 0.063±0.239 | **0.226±0.429** | 0.210±0.357 | 0.041±0.223 |
| POM | 3.359±1.833 | **4.869±1.202** | 4.001±1.602 | 4.175±0.745 | 3.952±0.612 | 4.985±0.590 | **4.993±0.843** | 3.533±1.203 |

*and efficient to serve as a fundamental safeguard against population degradation. Consequently, the best practice for parameterizing the selection operator is better NOT parameterizing it.*

### 4.3. Time Budget

We compare the training time costs of `MetaDistill` and MetaGBT for training POM over 128 epochs, as reported in Table 4. The results indicate that, excluding the additional overhead of knowledge base construction in `MetaDistill`, the training costs of the two frameworks are comparable. Even when this overhead is included (using the same knowledge base size as in Section 4.1), the increase in overall cost remains relatively modest. More importantly, the cost of constructing the knowledge base does not scale linearly with its size, as the knowledge base can be reused across multiple training runs. Additional construction costs are incurred only when introducing new teachers or new tasks.

*Table 4.* GPU hours for training POM over 128 epochs. The knowledge base contains 80 trajectories.

| Framework | GPU hours (RTX 5090) |
|---|---|
| MetaGBT | 0.1600 |
| `MetaDistill` (w/o constructing knowledge base) | 0.2195 |
| Constructing knowledge base | 0.0378 |

### 4.4. Parametric Analysis

**Goal** We investigate how the diversity of distillation data influences the generalization of `MetaDistill`. Diversity is characterized across two primary axes: (1) **Task Diversity** (`fsetK*`), where $K \in \{1, 3, 5, 10, 16\}$ denotes the number

of training tasks; and (2) **Teacher Diversity** (`tsetK*`), where $K \in \{1, 3, 7\}$ denotes the number of distinct teacher optimizers used to generate distillation trajectories (with the training task set fixed at 16). This systematic analysis allows us to isolate the impact of data diversity on the optimizers' ability to generalize to unseen objective functions.

**Results** Table 3 reports the logarithmic absolute improvement, measured by LAD, of `MetaDistill` variants over their respective baselines across various `fsetK` and `tsetK` configurations at $d = 100$. Detailed experimental settings and extended results are available in Appendix C. Our primary findings are summarized below:

- **More tasks often help, up to a point.** While increasing the training task count typically boosts performance, the effect varies by architecture. LGA exhibits monotonic improvements up to `fsetK16`, whereas LDE achieves performance saturation at `fsetK5`.

- **Diversity Paradox** Counter-intuitively, a larger ensemble of teachers can be counterproductive. For instance, POM's performance degrades when transitioning from `tsetK3` to `tsetK7`, and LES favors `tsetK1` at higher dimensions. We infer that since both LES and the CMA-ES teacher are based on Gaussian strategies, their distribution parameters $(\boldsymbol{\mu}, \boldsymbol{\Sigma})$ can be directly extracted from the optimizer states. This direct correspondence provides a more explicit and consistent loss signal for distillation than population-based estimation.

- **Optimal diversity is optimizer-specific.** LGA thrives on high task diversity, LDE prefers moderate task variety, and POM reaches peak performance with a balanced, moderate diversity of both tasks and teachers. In contrast, LES transitions from favoring task diversity at low

dimensions to preferring single-teacher training at high dimensions.

## 5. Conclusion

In this paper, we introduced `MetaDistill`, a versatile and effective framework designed to elevate the performance ceiling of learnable black-box optimizers. By leveraging an external knowledge base as a distillation source and employing Diversity-Preserving Distillation, we enable learnable optimizers to achieve significant performance gains across a wide range of challenging benchmarks. Furthermore, the integration of Self-Supervised Fine-Tuning during test-time allows the `MetaDistill` variants to further enhance their search efficiency and robustness in complex environments. Collectively, DPD and SSFT establish `MetaDistill` as a powerful paradigm for developing high-performance, learnable black-box optimizers.

Despite these advancements, several open questions remain. In particular, the intricate quantitative relationships among the performance of learnable optimizers, computational budget, and the specific configurations of teacher optimizers and training tasks (e.g., the tuning budget allocated to teacher optimizers and the diversity of both teacher optimizers and tasks) deserve further investigation. Moreover, the self-optimization of learnable optimizers represents another promising avenue for research. We identify these topics as important directions for future work.

## Acknowledgements

This work was supported in part by Natural Science Basic Research Program of Shaanxi under Grant 2025JC-QYCX-060, in part by the National Natural Science Foundation of China under Grant 62471371, 62472345, and U25B2018, in part by the Fundamental Research Funds for the Central Universities under Grant QTZX26124 and QTZX26119, and in part by the Open Research Fund of The State Key Laboratory of Blockchain and Data Security, Zhejiang University.

## Impact Statement

This paper presents work whose goal is to advance the field of Black-Box Optimization. There are many potential societal consequences of our work, none of which we feel must be specifically highlighted here.

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

# A. Parameters and Training Task Set

The parameter settings for all teacher optimizers used to generate distillation trajectories in `MetaDistill` are shown in Table 5.

The expressions for all the training functions used by `MetaDistill` are shown in Table 6.

*Table 5.* Detailed parameter settings for all teacher optimizers.

| Algorithm | item | setting |
|---|---|---|
| Shared settings | Problem setup | All teacher runs use $d=10$, population size $N=200$, and $G=100$ generations with minimization objective. |
| | Fitness evaluations | All algorithms are run with explicit box constraints given by the task definition (i.e., $[x_{\text{lb}}, x_{\text{ub}}]$), and population fitness is re-evaluated every generation. |
| CMA-ES | Initialization | Initial mean $\mathbf{m}_0$ is sampled i.i.d. from $\mathcal{U}(0, 1)$, and the initial step-size $\sigma_0 \sim \mathcal{U}(0, 1)$ (randomized per run). |
| | Bounds | Box constraints are set to the function domain $[x_{\text{lb}}, x_{\text{ub}}]$ (per dimension). |
| | Iterations | `maxiter = 100`, `popsize = 200`, and `verb_disp = 0`. |
| | Teacher distribution parameters | We use the internal CMA-ES mean $\mathbf{m}$ and covariance $\sigma\mathbf{C}$ as the teacher distribution parameters $(\boldsymbol{\mu}, \boldsymbol{\Sigma})$. |
| PSO | Hyperparameters | Global-best PSO with $w = 0.7298$, $c_1 = c_2 = 1.49618$, $v_{\max} = 0.2 \cdot (x_{\text{ub}} - x_{\text{lb}})$, $v_{\text{init}} = 0.1 \cdot (x_{\text{ub}} - x_{\text{lb}})$, and zero-velocity on clipping. |
| | Fitness handling | `assume_fitness = True`. |
| GA | Operators | Tournament selection ($k = 3$), SBX crossover ($p_c = 0.9$, $\eta_c = 15$), and polynomial mutation ($p_m = 1/d$, $\eta_m = 20$). |
| | Fitness handling | `assume_fitness = True`. |
| DE | Mutation factor | $F = 0.5$. |
| | Crossover rate | $CR = 0.5$ (binomial crossover). |
| JADE | Control parameters | $c = 0.1$, $p = 0.05$, $\mu_F = 0.5$, and $\mu_{CR} = 0.5$. |
| | Sampling | $F \sim \text{Cauchy}(\mu_F, 0.1)$ (truncated to $(0, 1]$) and $CR \sim \mathcal{N}(\mu_{CR}, 0.1)$ (clipped to $[0, 1]$). |
| SHADE | Control parameters | $p = 0.05$, memory size $H = 20$, and archive size $|A| = N$ ($N = 200$). |
| | Initialization | The success-history memories are initialized as $M_F = M_{CR} = 0.5$. |
| L-SHADE | Control parameters | $p = 0.05$, memory size $H = 20$, $NP_{\min} = 4$, and archive factor $1.0$. |
| | Budget for LPSR | `max_fe = N · G = 200 × 100 = 20000` (used for linear population size reduction). |
| | Initialization | The success-history memories are initialized as $M_F = M_{CR} = 0.5$. |

*Table 6.* All training tasks for `MetaDistill`

| ID | Expressions | Features |
|---|---|---|
| TF1 | $\sum_i^d |x_i - b_i|$ | $\mathbf{x}, \mathbf{b} \in [-10, 10]^d$ |
| TF2 | $\sum_i^{d-1} |(x_i - b_i) + (x_{i+1} - b_{i+1})| + \sum_i^d |x_i - b_i|$ | $\mathbf{x}, \mathbf{b} \in [-10, 10]^d$ |
| TF3 | $10^{-4} \times \sum_i^d (x_i - b_i)^2$ | $\mathbf{x}, \mathbf{b} \in [-100, 100]^d$ |
| TF4 | $0.1 \times \max \{|x_i - b_i|, 1 \leq i \leq d\}$ | $\mathbf{x}, \mathbf{b} \in [-100, 100]^d$ |
| TF5 | $10^{-9} \times \sum_{i=1}^{d-1} \left[ 100 \times \left( (x_i - b_i)^2 - (x_{i+1} - b_{i+1}) \right)^2 + (x_i - b_i - 1)^2 \right]$ | $\mathbf{x}, \mathbf{b} \in [-100, 100]^d$ |
| TF6 | $\frac{200}{d} \times \sum_i^d \cos(|x_i - b_i|)$ | $\mathbf{x}, \mathbf{b} \in [-10, 10]^d$ |
| TF7 | $\sum_i^d sin(x_i - b_i)$ | $\mathbf{x}, \mathbf{b} \in [-10, 10]^d$ |
| TF8 | $\sum_i^d \sqrt{|x_i - b_i|}$ | $\mathbf{x}, \mathbf{b} \in [-100, 100]^d$ |
| TF9 | $\ln \left( 2 + \frac{1}{d} \times \sum_i^d |x_i - b_i| \right)$ | $\mathbf{x}, \mathbf{b} \in [-100, 100]^d$ |
| TF10 | $\frac{10^{-3}}{d} \times \sum_i^d |(x_i - b_i)^3|$ | $\mathbf{x}, \mathbf{b} \in [-10, 10]^d$ |
| TF11 | $2^{\frac{0.1}{d} \times \sum_i^d |x_i - b_i|}$ | $\mathbf{x}, \mathbf{b} \in [-50, 50]^d$ |
| TF12 | $\sum_i^d \left[ (x_i - b_i)^2 - (x_i - b_i) \right]$ | $\mathbf{x}, \mathbf{b} \in [-15, 15]^d$ |
| TF13 | $\frac{1}{5} \times \sum_{i=1}^5 |x_i - b_i| + \frac{1}{d-5} \times \sum_{j=6}^d |x_j - b_j|$ | $\mathbf{x}, \mathbf{b} \in [-100, 100]^d$ |
| TF14 | $\frac{2}{d} \times \sum_{i=1}^{d/2} \left[ |x_i - b_i| + |x_{i+d/2} - b_{i+d/2}| \right] + \frac{1}{d} \times \sum_{i=1}^d |x_i - b_i|$ | $\mathbf{x}, \mathbf{b} \in [-100, 100]^d$ |
| TF15 | $\sum_{i=1}^2 (x_i - b_i)^2 + \sum_{j=3}^d (x_j - b_j)^2$ | $\mathbf{x}, \mathbf{b} \in [-10, 10]^d$ |
| TF16 | $\frac{25}{d} \sum_{i=1}^d (\sin (x_i - b_i) + \cos (x_i - b_i))$ | $\mathbf{x}, \mathbf{b} \in [-100, 100]^d$ |

# B. Additional Results for BBOB

## B.1. BBOB Evaluation

**Benchmark suite and dimensions.**    We evaluate generalization on the BBOB test suite (Hansen et al., 2021), including all 24 functions ($f_1$–$f_{24}$). Unless otherwise stated, we report results for dimensions $d \in \{30, 100, 200, 300\}$ in the text.

**Evaluation protocol.**    For every (optimizer variant, function, dimension) setting, we run the optimizer with a fixed population size $N{=}200$ and a budget of $B{=}10{,}000$ function evaluations. The search domain is fixed to $[-5, 5]^d$ for all BBOB evaluations. No restarts are employed.

**Seeds and aggregation.**    Unless otherwise stated, we run 9 independent random seeds (seeds 0–8) per setting. To obtain robust estimates while keeping the evaluation protocol consistent across all optimizers, we aggregate results using three overlapping 7-seed windows: seeds 0–6, 1–7, and 2–8, and report their average (denoted as `Averaged` in our figures and tables). For each seed-window evaluation, the reported *final objective value* is the last point of the mean convergence curve (mean over seeds in the window), and Table 1 averages this final value over $f_1$–$f_{24}$ for each dimension.

**Fixed BBOB instances (offsets).**    To ensure fair comparisons, we fix the BBOB instance parameters across all variants and seeds by using pre-generated offsets.

**Convergence metric.**    We use the `best-gen` metric (best-of-generation): at each generation we record the minimum objective value among the current population. We plot convergence curves as $\log_{10}$ of this `best-gen` fitness (with a numerically safe shift if needed).

**Variants and SSFT.**    We compare each optimizer's baseline against `MetaDistill` variants: `md_j0`, `md_j1`, `md_j3`, and `md_j5`. The `md_j*` variants enable test-time self-supervised fine-tuning with self-supervision interval $j \in \{1, 3, 5\}$, while `baseline` and `md_j0` are evaluated without adaptation. Learning rate is fixed to $10^{-4}$ across all four optimizers. Crucially, the self-supervised interval carries distinct semantics depending on the optimization context. In self-supervised objectives, it defines the lookback step size for adaptive signals, with parameter updates occurring at every generation. Conversely, in black-box objectives, it denotes the actual update frequency. Since these two objective types are mutually exclusive in our framework, we adopt a unified notation for both to maintain conceptual simplicity.

## B.2. Additional BBOB Convergence Curves

Figure 5–20 shows the $\log_{10}$ convergence curves of baselines and `MetaDistill` variants of each learnable optimizer in dimension $d = 30, 100, 200$ and 300, where "Teacher-Synth" indicates the construction of the optimal trajectory using all teachers' trajectories collected on each test task in the same way as the training trajectories, representing the optimal level of all teachers.

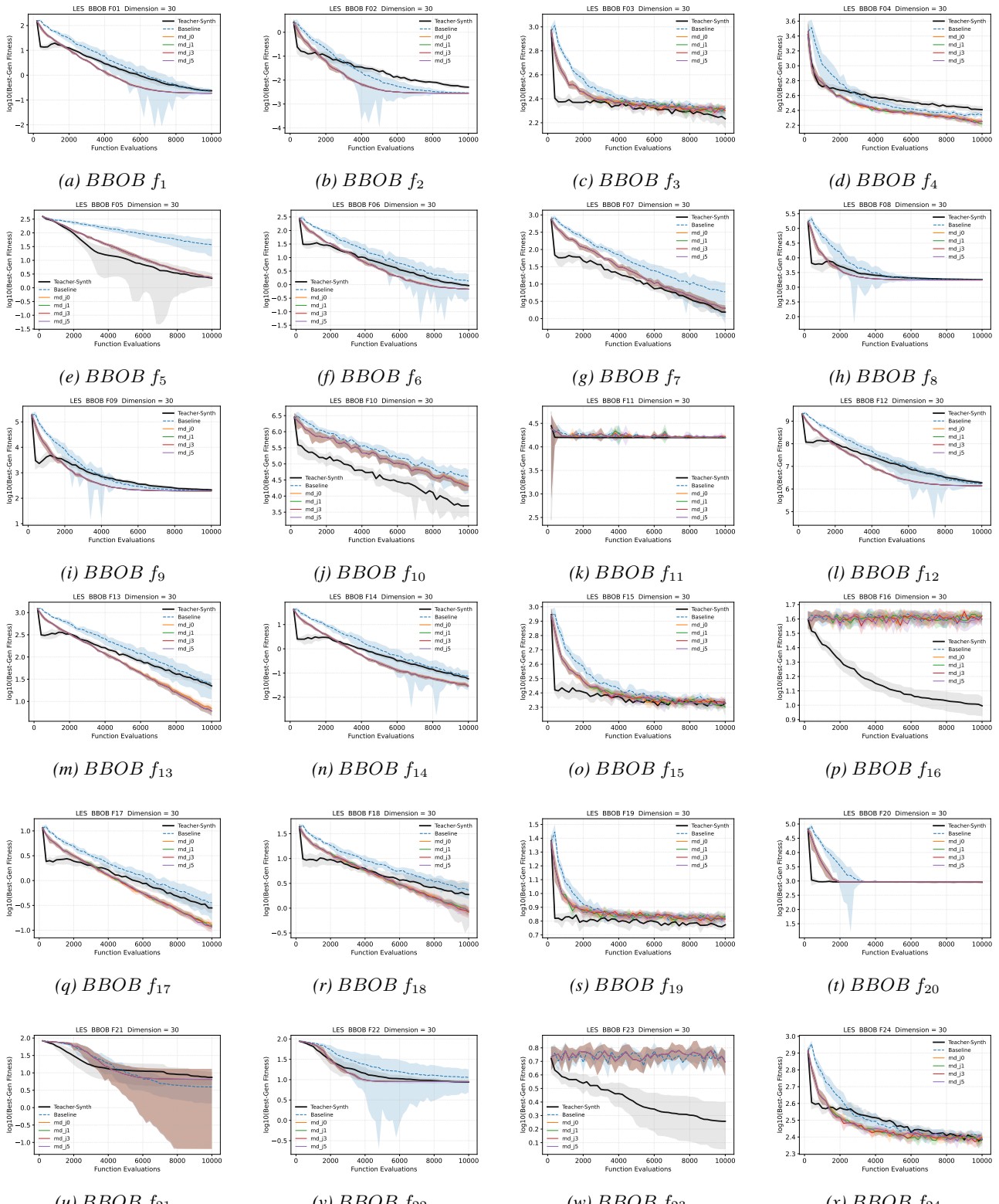

*Figure 5.* Log convergence curves of LES on $BBOB$ $f_1 \sim f_{24}$, Dimension = 30.

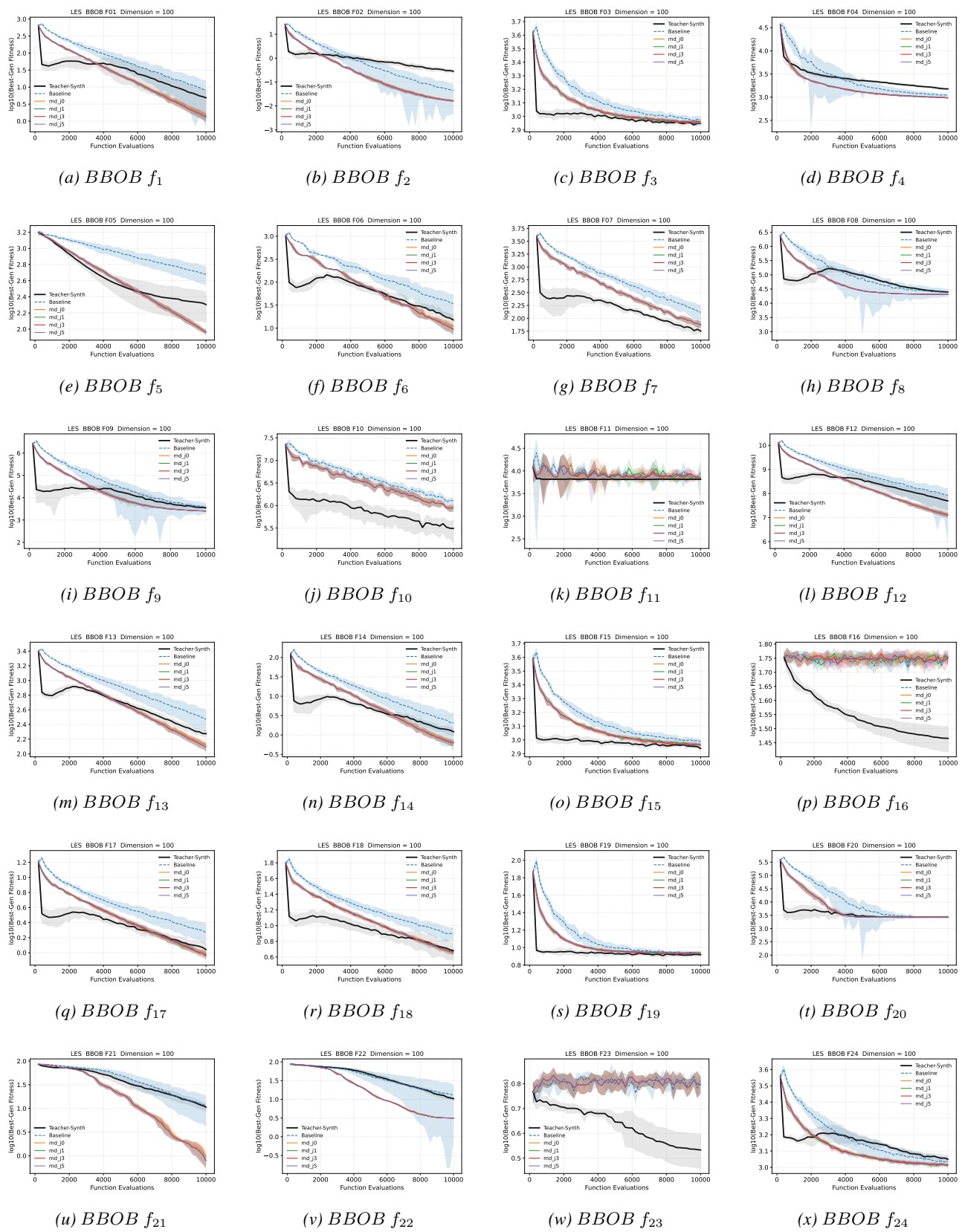

*Figure 6.* Log convergence curves of LES on $BBOB$ $f_1 \sim f_{24}$, Dimension = 100.

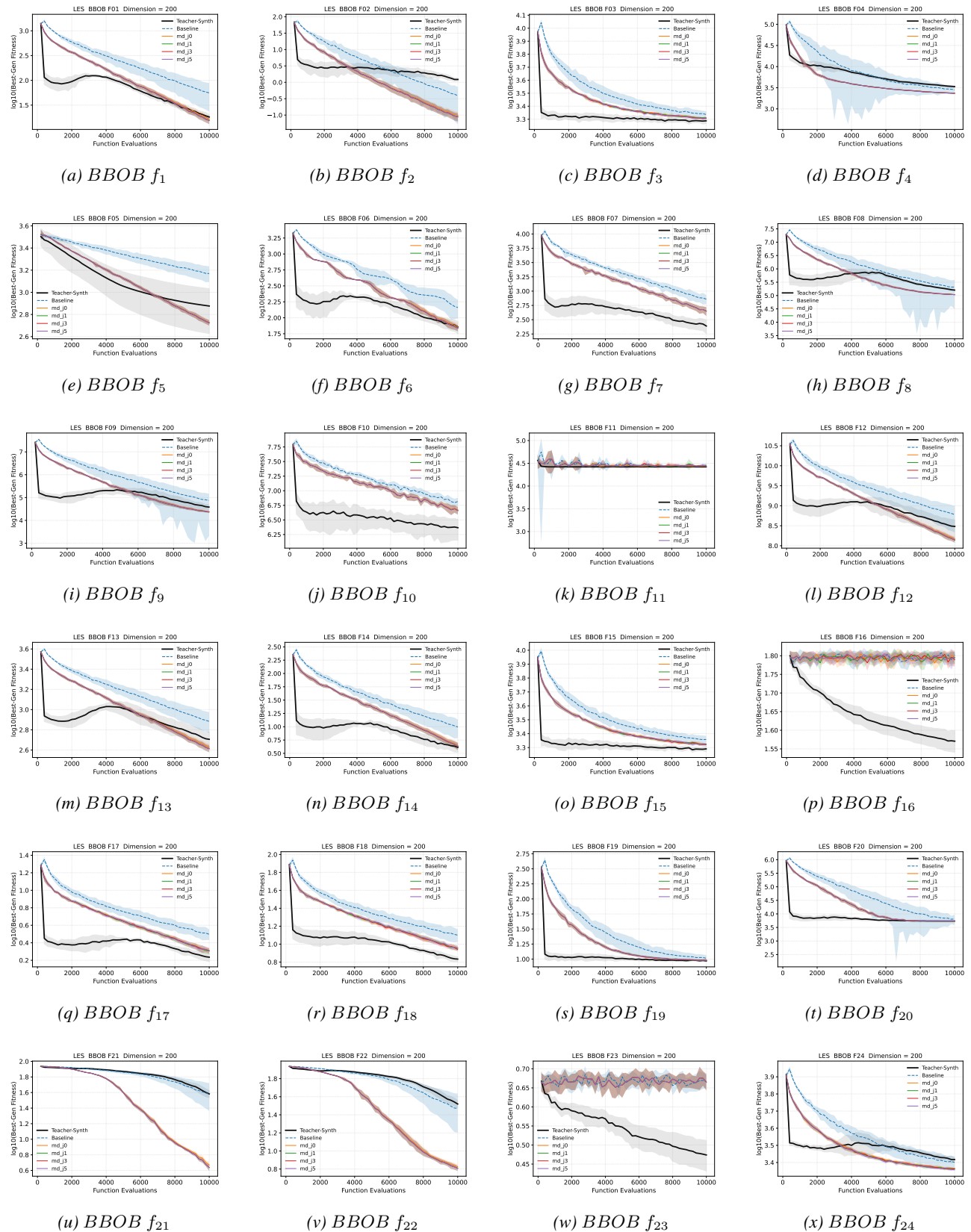

*Figure 7.* Log convergence curves of LES on $BBOB$ $f_1 \sim f_{24}$, Dimension = 200.

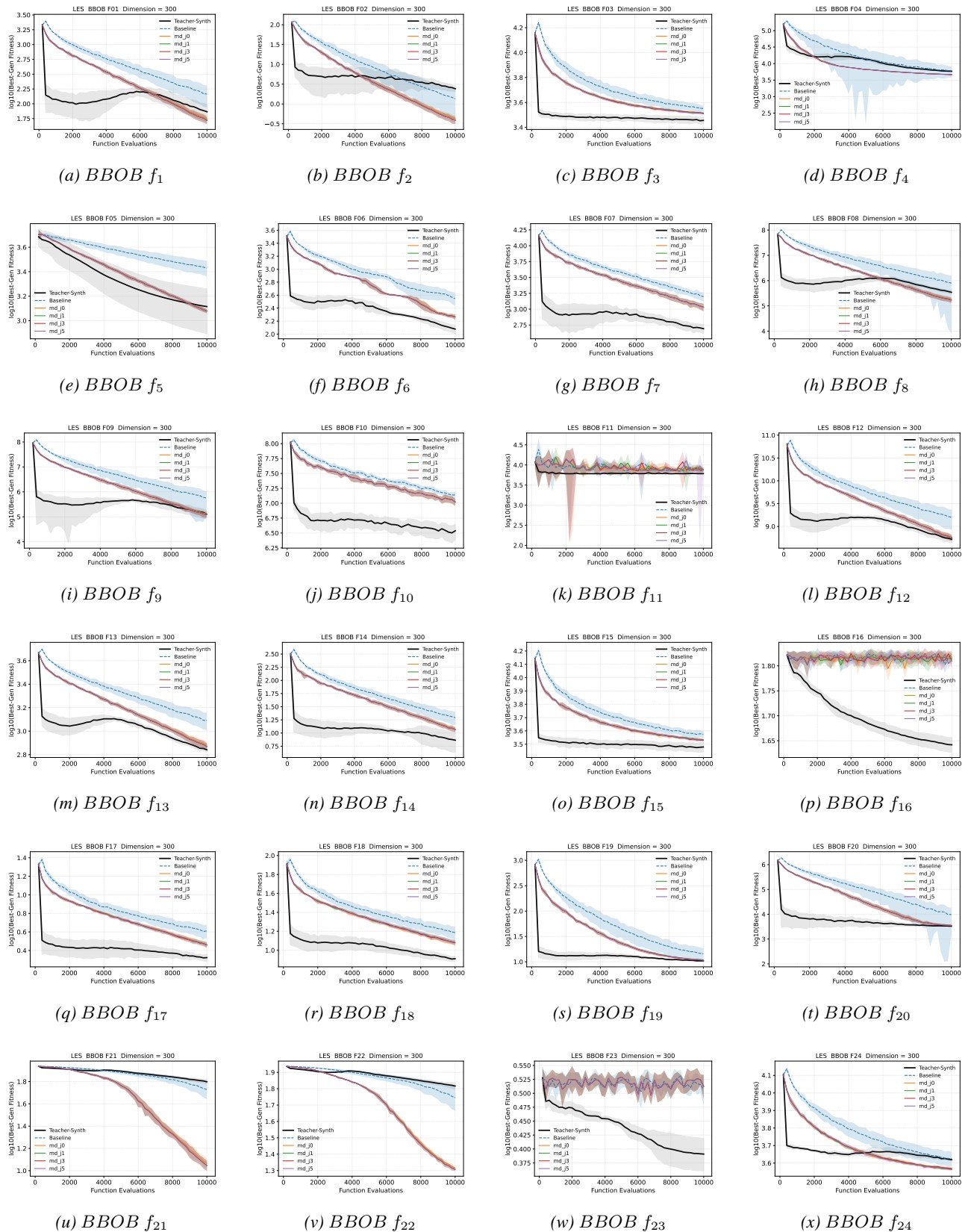

*(a) BBOB $f_1$*  *(b) BBOB $f_2$*  *(c) BBOB $f_3$*  *(d) BBOB $f_4$*

*(e) BBOB $f_5$*  *(f) BBOB $f_6$*  *(g) BBOB $f_7$*  *(h) BBOB $f_8$*

*(i) BBOB $f_9$*  *(j) BBOB $f_{10}$*  *(k) BBOB $f_{11}$*  *(l) BBOB $f_{12}$*

*(m) BBOB $f_{13}$*  *(n) BBOB $f_{14}$*  *(o) BBOB $f_{15}$*  *(p) BBOB $f_{16}$*

*(q) BBOB $f_{17}$*  *(r) BBOB $f_{18}$*  *(s) BBOB $f_{19}$*  *(t) BBOB $f_{20}$*

*(u) BBOB $f_{21}$*  *(v) BBOB $f_{22}$*  *(w) BBOB $f_{23}$*  *(x) BBOB $f_{24}$*

*Figure 8.* Log convergence curves of LES on $BBOB$ $f_1 \sim f_{24}$, Dimension = 300.

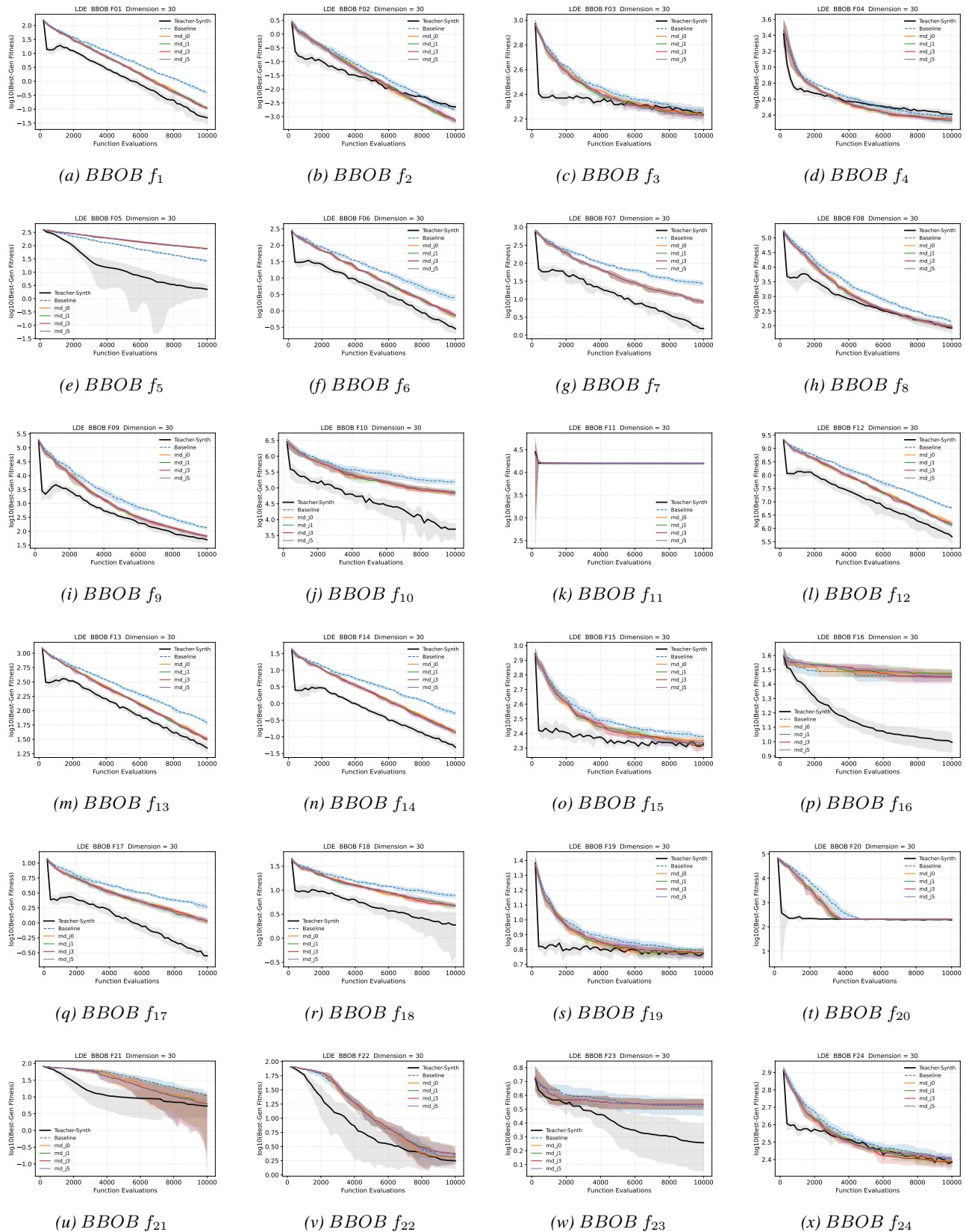

*Figure 9.* Log convergence curves of LDE on $BBOB\ f_1 \sim f_{24}$, Dimension = 30.

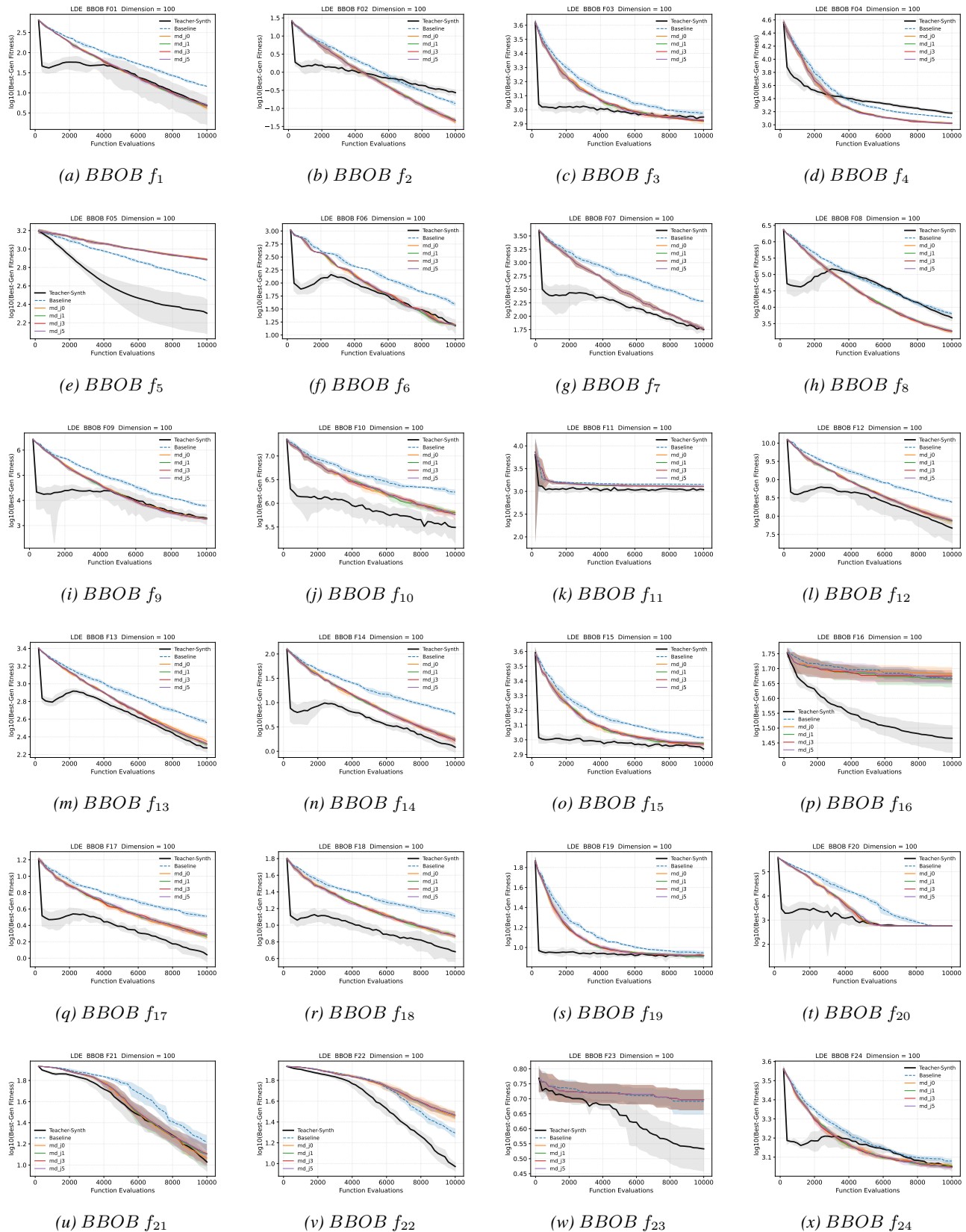

*Figure 10.* Log convergence curves of LDE on $BBOB$ $f_1 \sim f_{24}$, Dimension = 100.

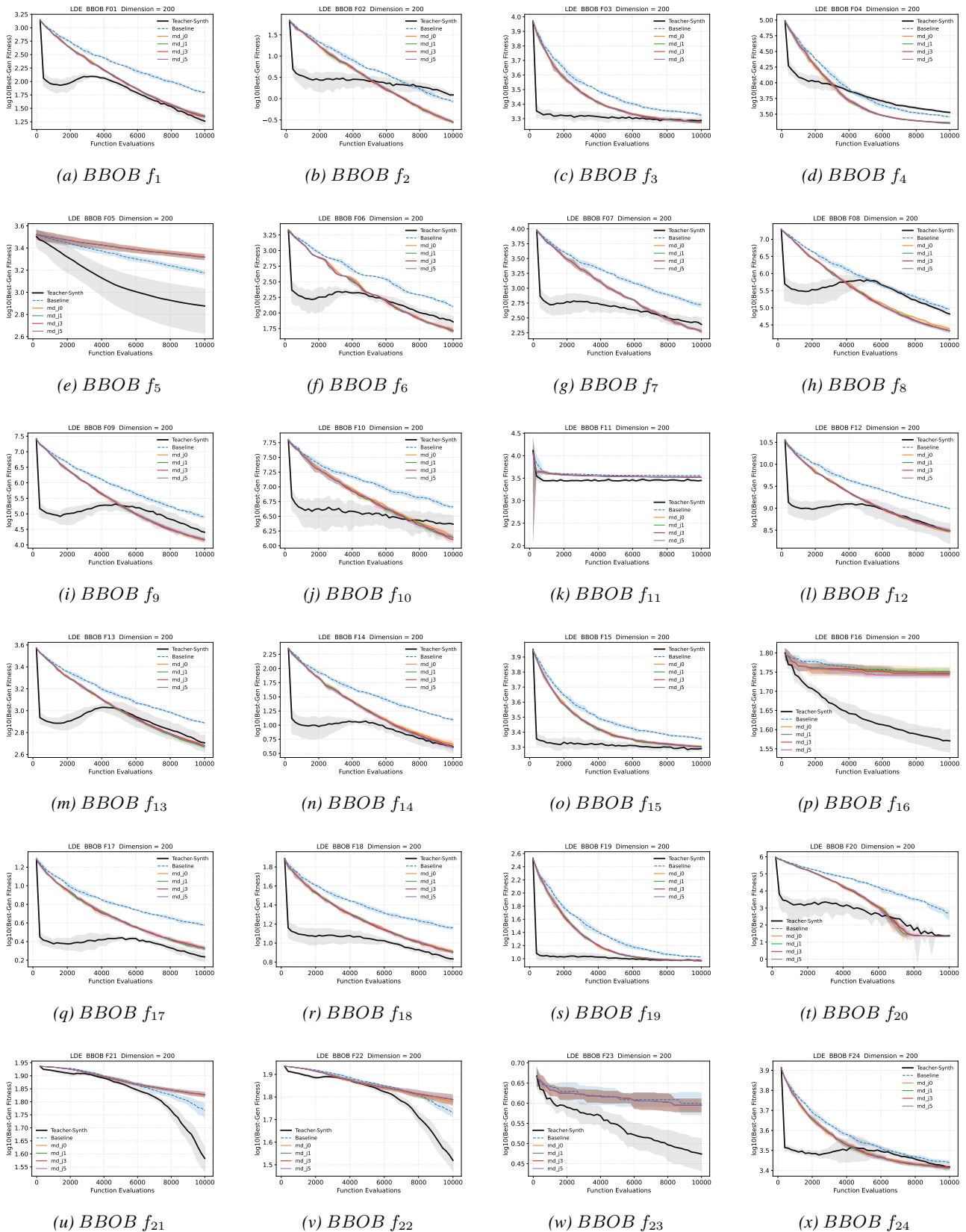

*Figure 11.* Log convergence curves of LDE on $BBOB\ f_1 \sim f_{24}$, Dimension = 200.

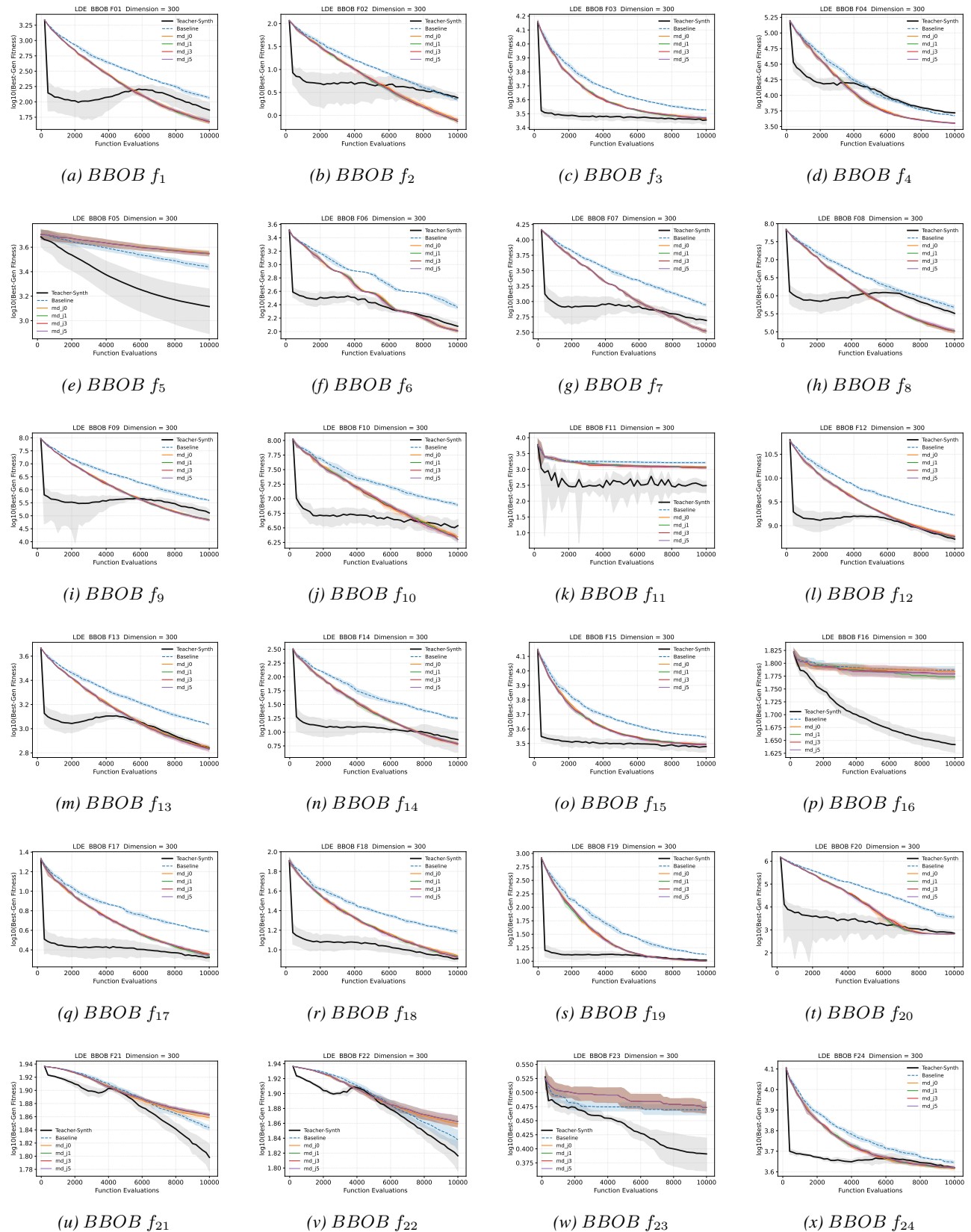

*Figure 12.* Log convergence curves of LDE on $BBOB$ $f_1 \sim f_{24}$, Dimension = 300.

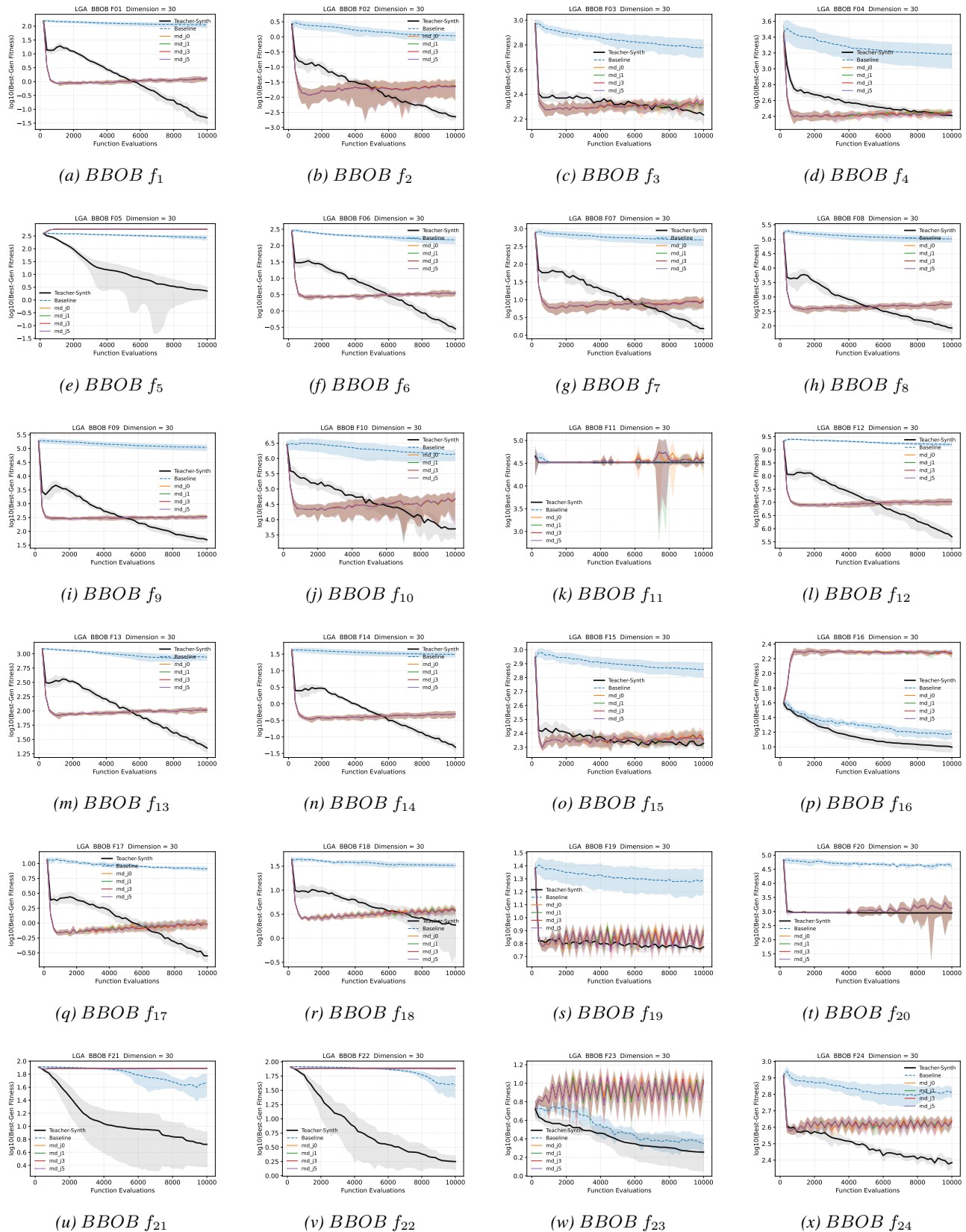

*Figure 13.* Log convergence curves of LGA on $BBOB\ f_1 \sim f_{24}$, Dimension = 30.

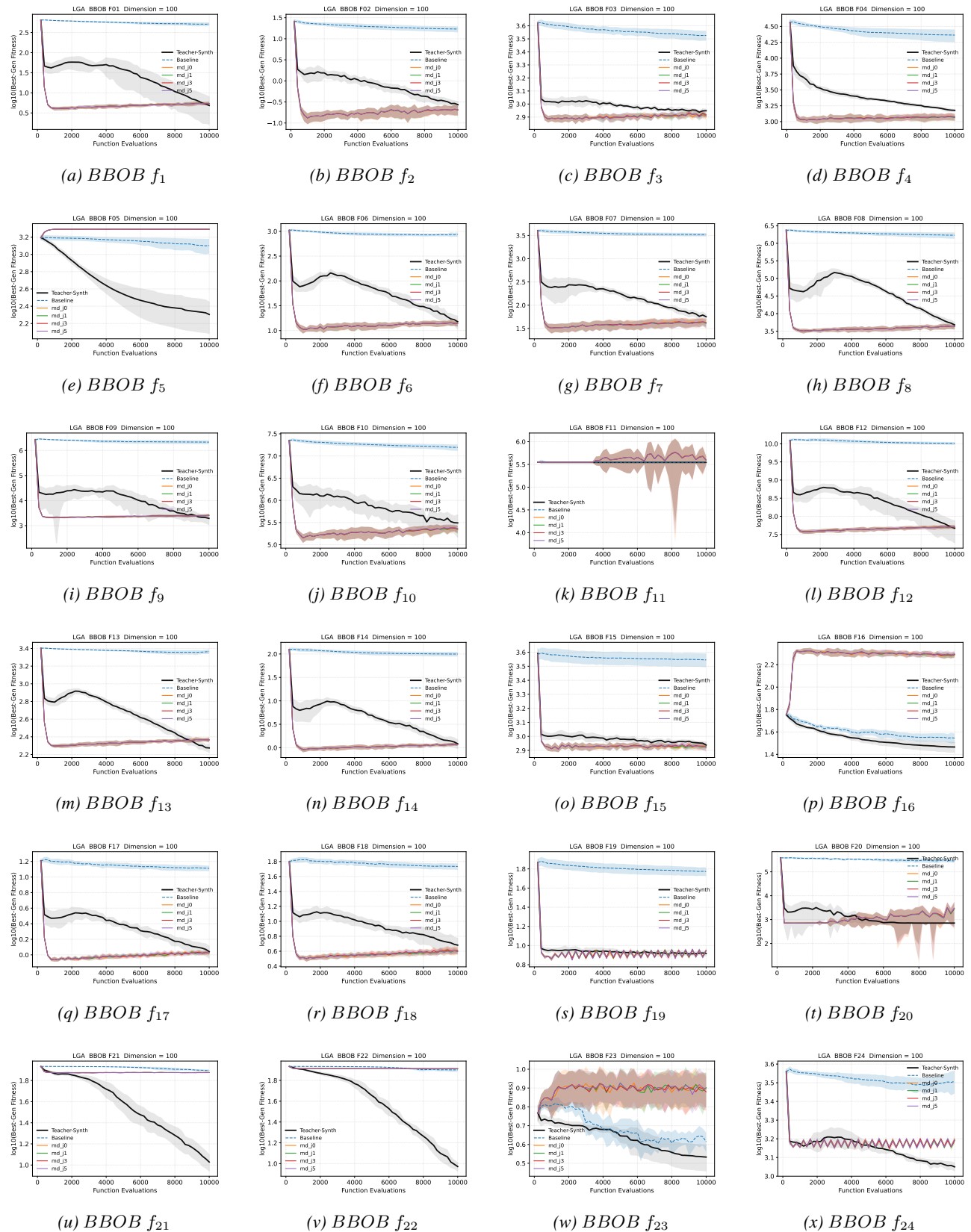

*Figure 14.* Log convergence curves of LGA on $BBOB$ $f_1 \sim f_{24}$, Dimension = 100.

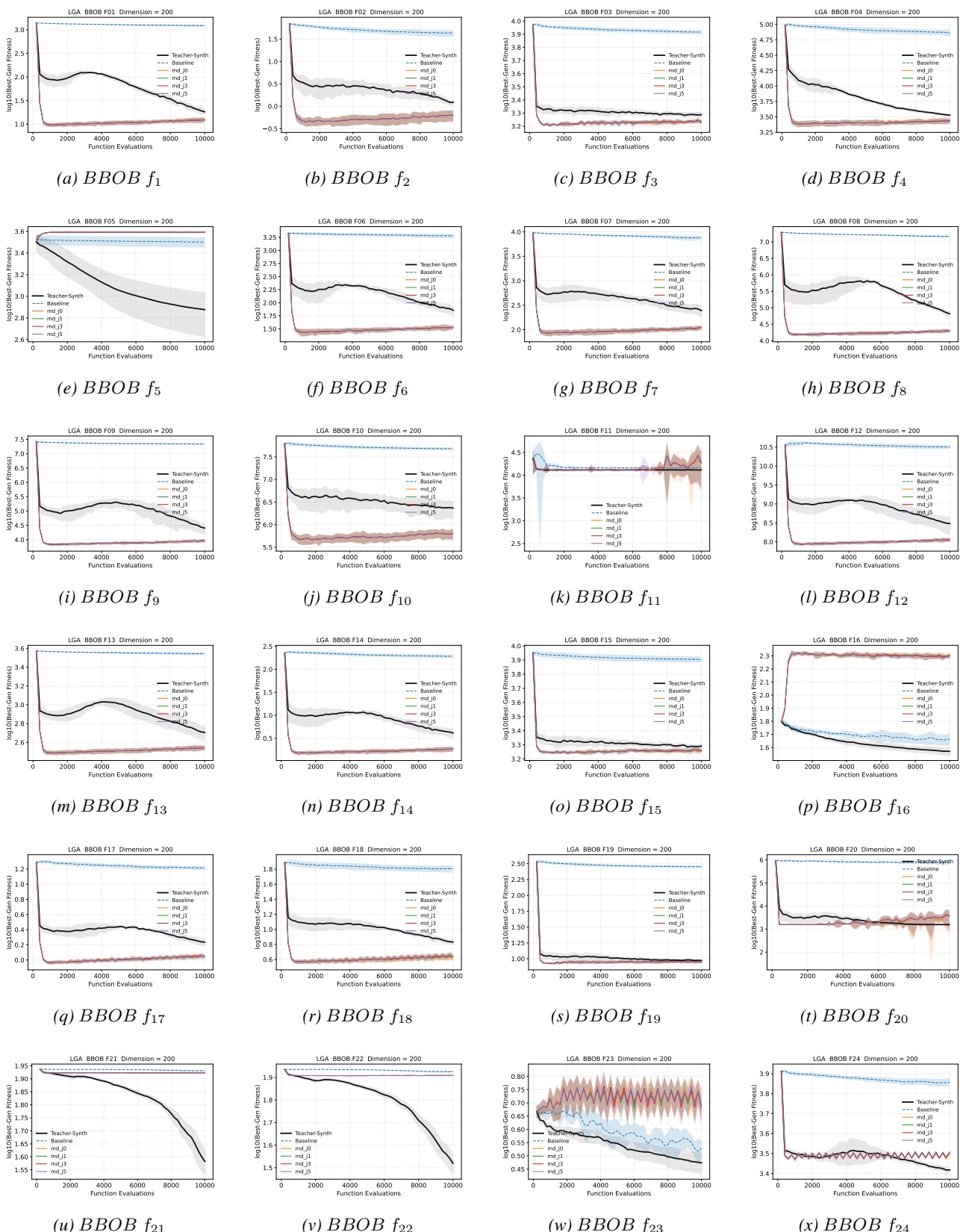

*Figure 15.* Log convergence curves of LGA on $BBOB$ $f_1 \sim f_{24}$, Dimension = 200.

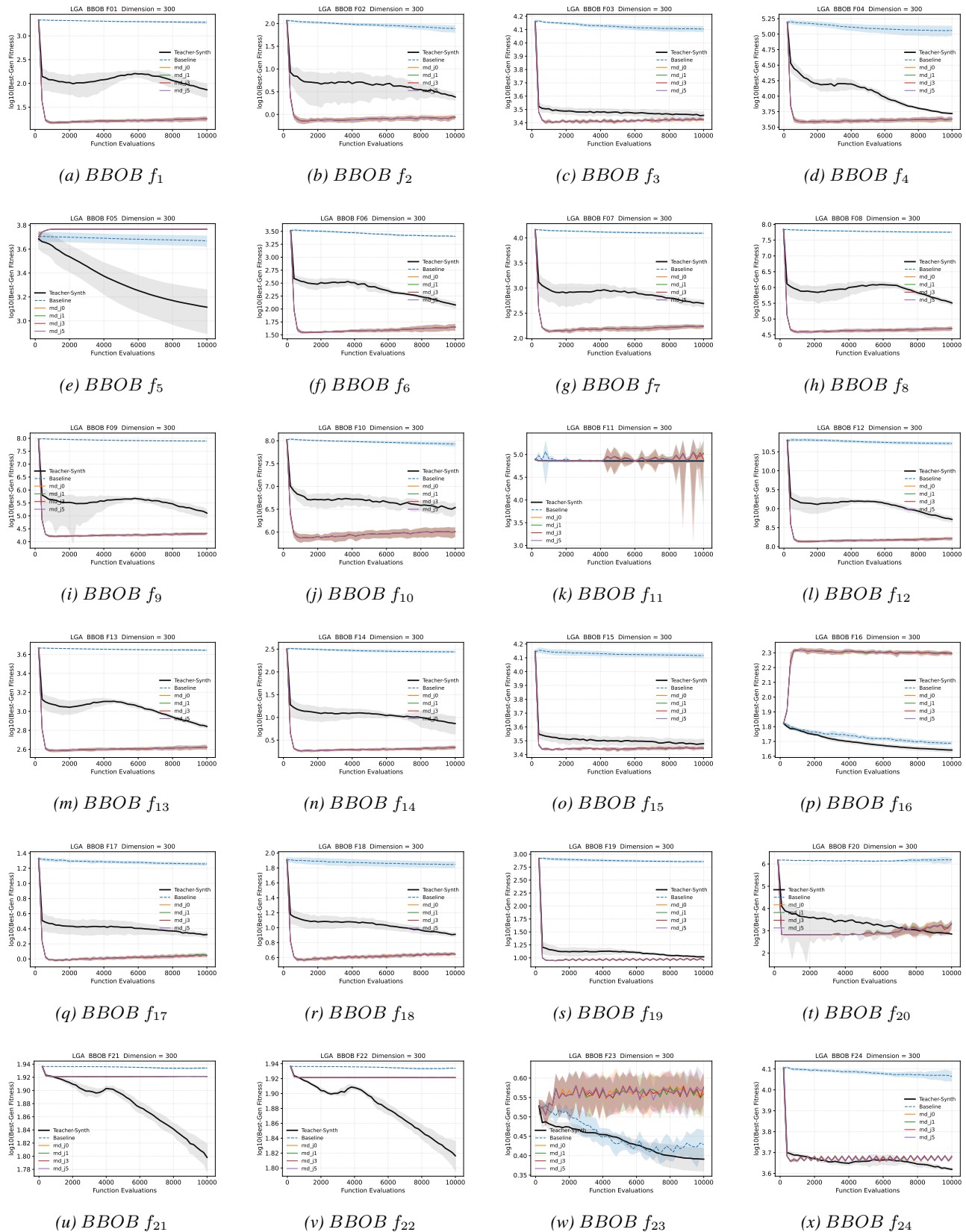

*Figure 16.* Log convergence curves of LGA on $BBOB$ $f_1 \sim f_{24}$, Dimension = 300.

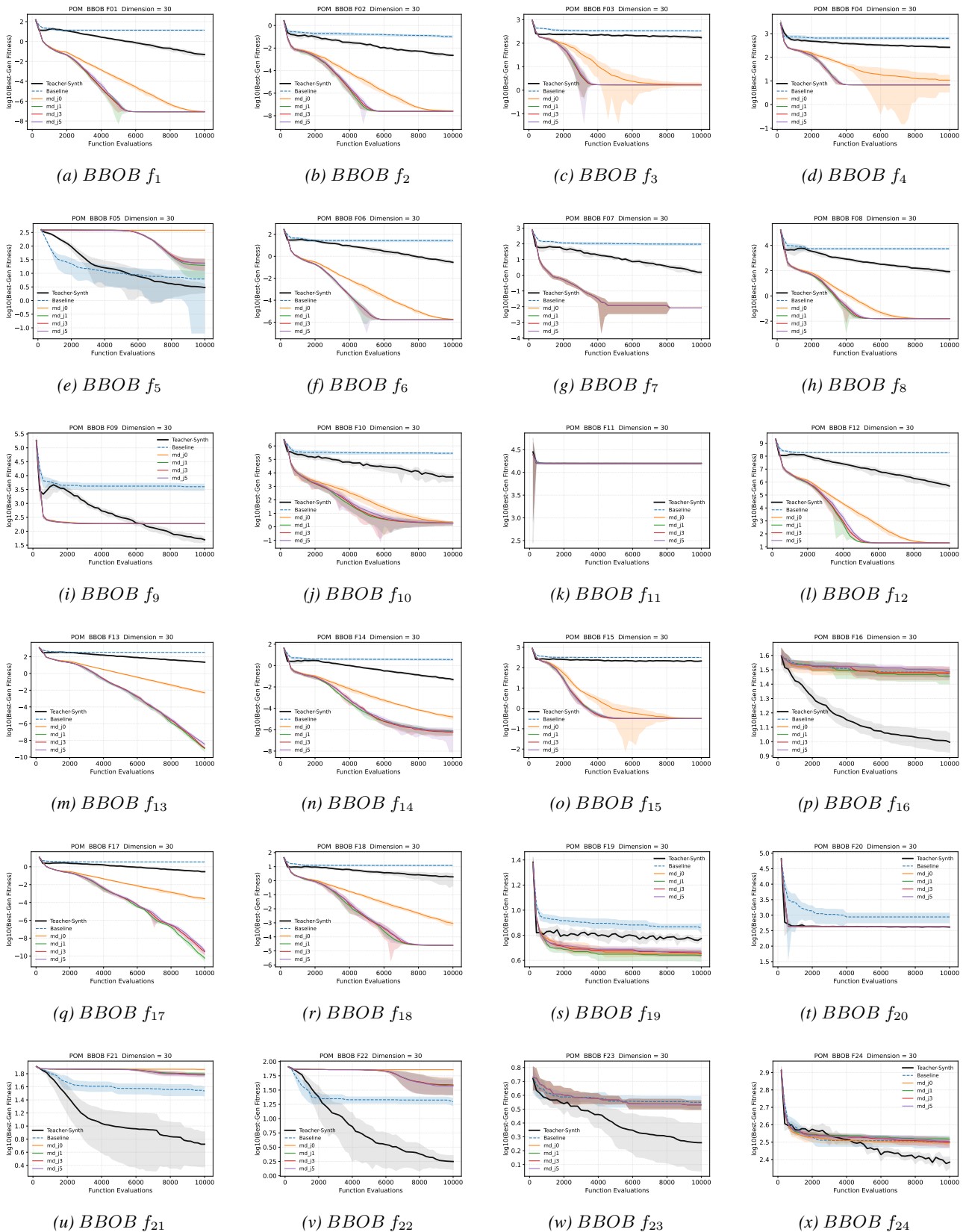

*Figure 17.* Log convergence curves of POM on $BBOB$ $f_1 \sim f_{24}$, Dimension = 30.

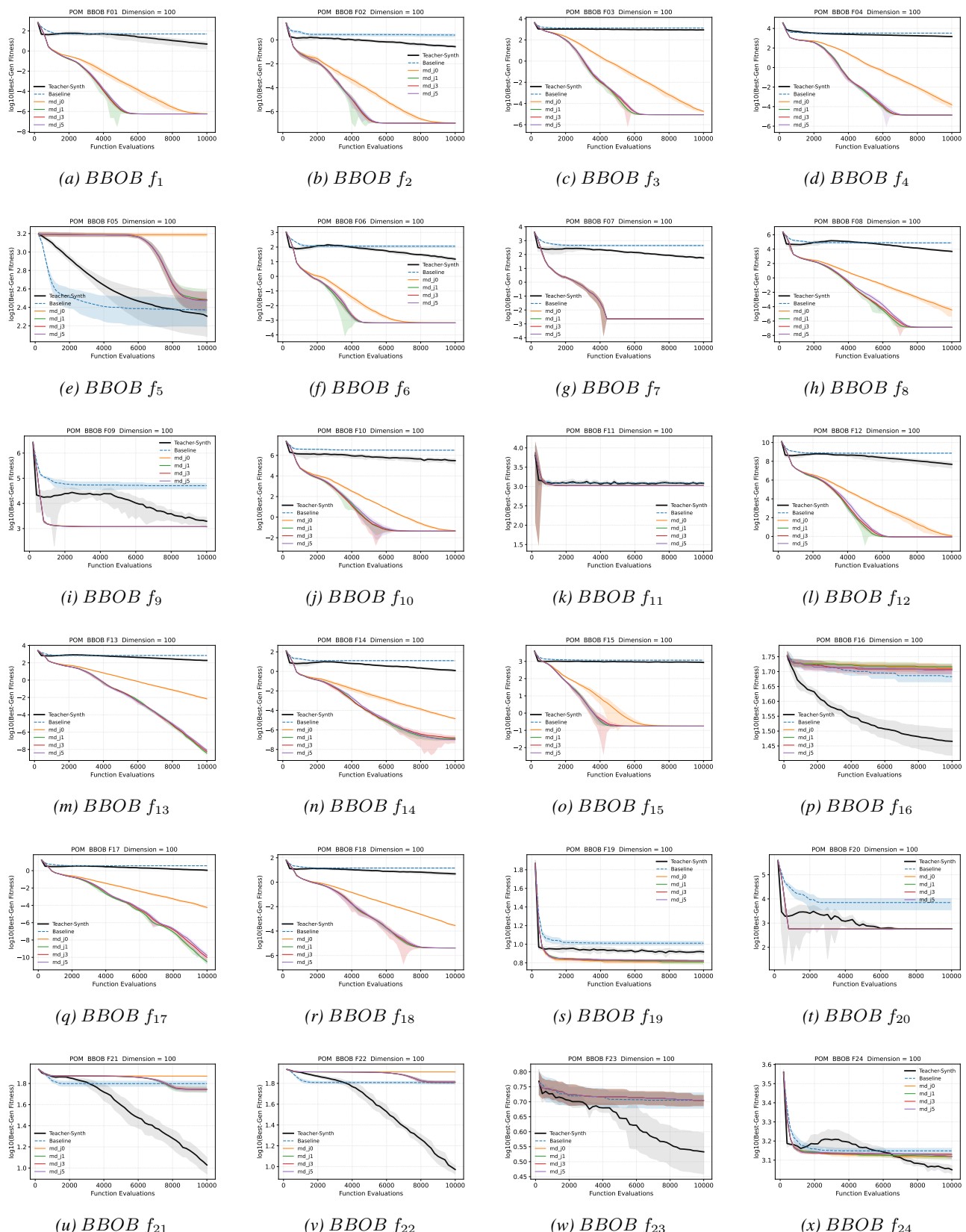

*Figure 18.* Log convergence curves of POM on $BBOB\ f_1 \sim f_{24}$, Dimension = 100.

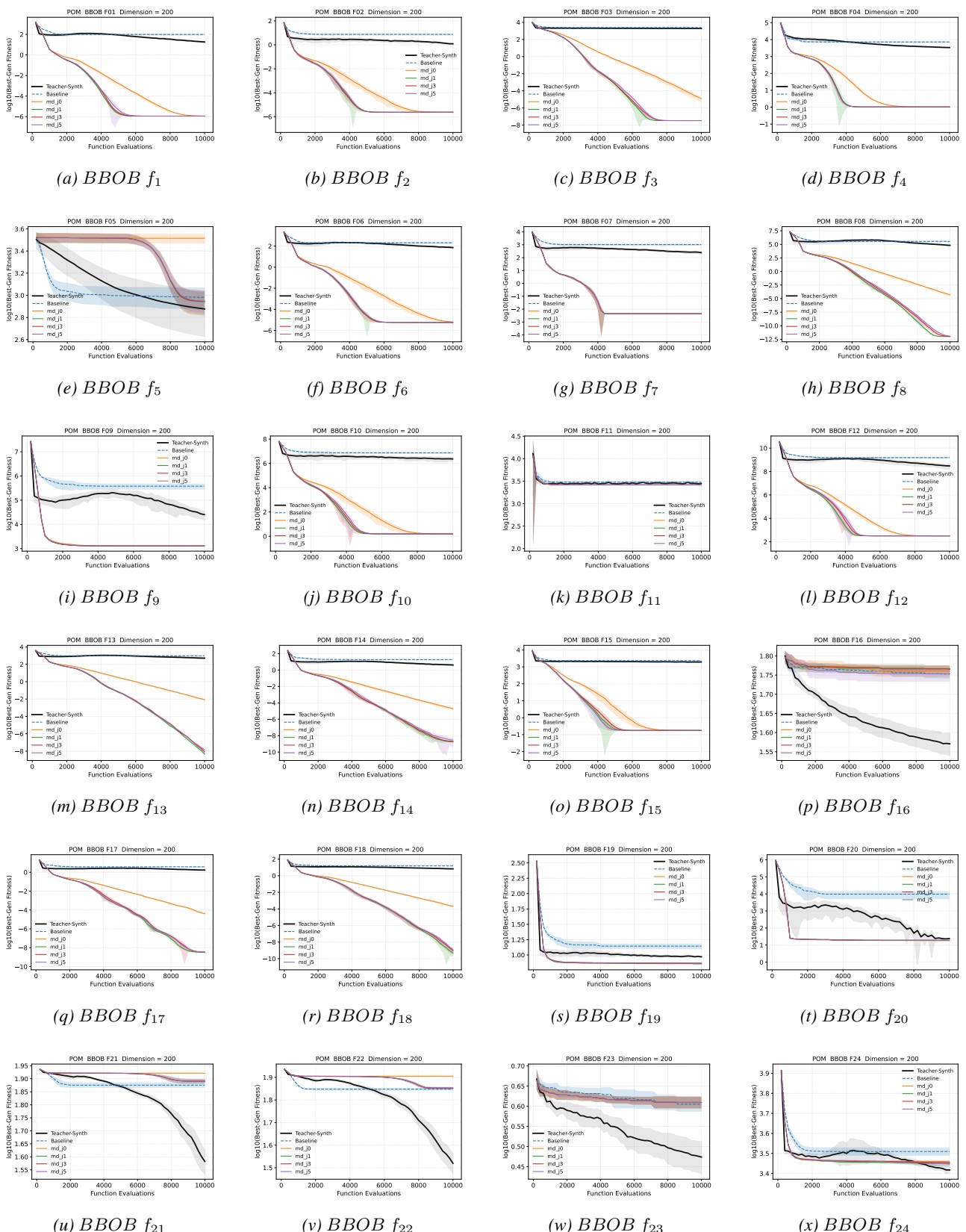

*Figure 19.* Log convergence curves of POM on $BBOB$ $f_1 \sim f_{24}$, Dimension = 200.

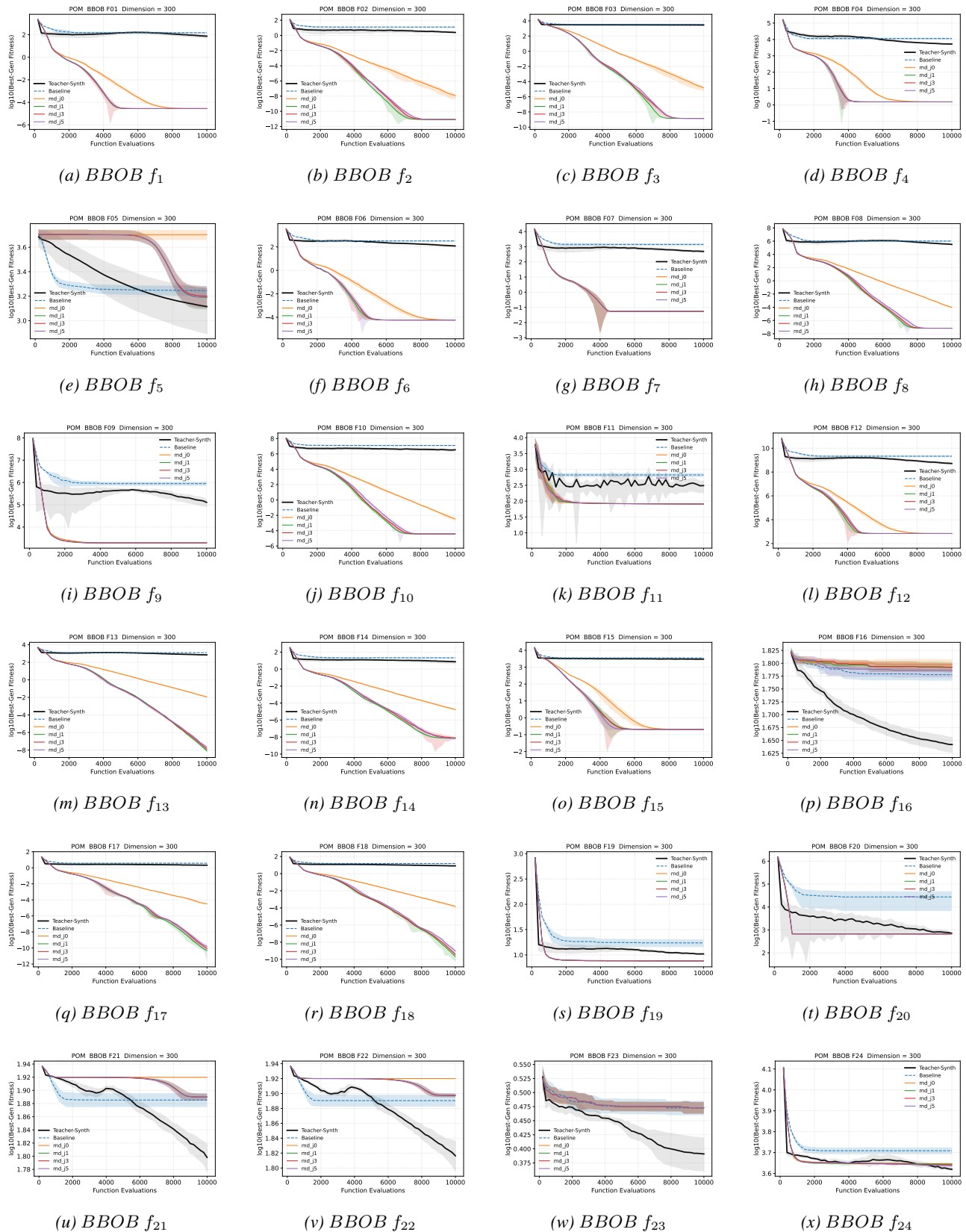

*Figure 20.* Log convergence curves of POM on $BBOB\ f_1 \sim f_{24}$, Dimension = 300.

## C. Additional Experimental Results for Parametric Analysis

### C.1. Metric definitions

For each BBOB function $f$ (indexed $1 \leq f \leq 24$) and dimension $d$, we evaluate each method with 8 independent evaluation seeds $s \in \{0, 1, \ldots, 7\}$ and record the final-generation objective values.[3] Let $y_{f,d,s}^{(\text{baseline})}$ denote the final value of the baseline run with eval seed $s$, and let $y_{f,d,r,s}^{(\text{md})}$ denote the final value of the MetaDistill variant (md) trained with train seed $r \in \{0, 1, \ldots, 7\}$ and evaluated with eval seed $s$. We define the per-function *final mean* by averaging over evaluation seeds:

$$Y_{f,d}^{(\text{baseline})} = \frac{1}{8} \sum_{s=0}^{7} y_{f,d,s}^{(\text{baseline})}, \qquad Y_{f,d,r}^{(\text{md})} = \frac{1}{8} \sum_{s=0}^{7} y_{f,d,r,s}^{(\text{md})}. \tag{7}$$

We then derive three aggregate metrics from these final means. First, **logabs** is the average $\log_{10}$ absolute final value across all 24 functions (lower is better):

$$\text{logabs}_d^{(\cdot)} = \frac{1}{24} \sum_{f=1}^{24} \log_{10}\Big(\big|Y_{f,d}^{(\cdot)}\big| + 10^{-12}\Big), \tag{8}$$

where $(\cdot)$ denotes either baseline (with $Y_{f,d}^{(\text{baseline})}$) or md (with $Y_{f,d,r}^{(\text{md})}$). Second, we define the log-absolute difference (LAD) per train seed $r$ as:

$$\text{LAD}_{d,r} = \text{logabs}_d^{(\text{baseline})} - \text{logabs}_{d,r}^{(\text{md})}. \tag{9}$$

A larger LAD indicates that the MetaDistill variant achieved a lower logabs (better final performance) than the baseline. Finally, **W/T/L** counts wins, ties, and losses (out of 24) when comparing md against the baseline per function: for each $f$, md is a win if $Y_{f,d,r}^{(\text{md})} < Y_{f,d}^{(\text{baseline})}$ (lower is better), a loss if it is larger, and a tie if the two values are nearly equal (we use a strict tolerance, implemented as `np.isclose` with `rtol=atol=`$10^{-12}$). We report W/T/L per train seed and aggregate over the 8 train seeds (mean $\pm$ standard deviation).

**Numerical stability.** BBOB objective values can vary drastically in magnitude across different functions and dimensions. Using a base-10 logarithm for final outcomes (as in the logabs metric) makes comparisons across such scales more interpretable: performance differences spanning several orders of magnitude are converted into differences on an additive $\log_{10}$ scale. The absolute value inside the log ensures the metric reflects the outcome's magnitude regardless of sign.[4] We include a small offset $\epsilon = 10^{-12}$ to avoid undefined values when $Y_{f,d} = 0$ and to prevent divergence for extremely small $|Y_{f,d}|$. If $|Y_{f,d}| \gg \epsilon$, adding this $\epsilon$ has negligible effect:

$$\log_{10}\Big(|Y_{f,d}| + \epsilon\Big) = \log_{10}\big(|Y_{f,d}|\big) + \log_{10}\Big(1 + \frac{\epsilon}{|Y_{f,d}|}\Big), \tag{10}$$

so the deviation from $\log_{10}(|Y_{f,d}|)$ is at most $\log_{10}\big(1 + \epsilon/|Y_{f,d}|\big)$, which is very small for large $|Y_{f,d}|$. Across all results reported in this work, the minimum observed $|Y_{f,d}|$ was 0, the median was $\approx 3.09 \times 10^2$, and the maximum was $\approx 4.71 \times 10^{12}$. Only about $0.61\%$ of all $Y_{f,d}$ values were below $10^{-11}$ in magnitude. Thus, the $10^{-12}$ offset safeguards numerical stability without materially changing the log-scale metrics (it only influences the $\log_{10}$ values in $\sim 0.6\%$ of nearly-zero cases).

**Interpretation.** The LAD metric is measured on a $\log_{10}$ scale, which aids interpretation of performance gains. For example, $\text{LAD} = 1$ corresponds to a $10\times$ improvement in the geometric-mean sense: since logabs is an average of $\log_{10}(\cdot)$ terms, a difference of 1 in logabs implies, on average across functions, a factor of $10^1$ in the underlying scale. The W/T/L metric, on the other hand, should be interpreted with caution as it is insensitive to the magnitude of improvement. If the md variant's advantage on a function grows or shrinks without flipping a win to a loss (or vice versa), the W/T/L tally remains unchanged—even a very small edge counts as a win, and a large improvement on a function already counted as a win does not increase the win count. Thus, W/T/L serves as a coarse indicator of per-function success. For a more nuanced comparison, one should examine complete performance profiles (e.g., the heatmaps in Figure 21 or convergence curves in Appendix C.5), which reveal the magnitude of improvement on each function.

---

[3]In our logs, both training and evaluation seeds use the range 0–7; we use distinct indices ($r$ vs. $s$) in the notation to avoid ambiguity.

[4]Due to fixed problem-specific offsets in BBOB, some final means $Y_{f,d}$ can be negative; using $|Y_{f,d}|$ guarantees the log input is non-negative.

*Table 7.* Parameter-study tags. `fsetK*` varies the number of training tasks ($K \in \{1, 3, 5, 10, 16\}$) using the full teacher pool. `tsetK*` fixes the training task set to 16 tasks and varies the number of teachers ($K \in \{1, 3, 7\}$).

| Tag | $K$ | Training task set |
|---|---|---|
| `fsetK1` | 1 | $TF_1$ |
| `fsetK3` | 3 | $TF_1$-$TF_3$ |
| `fsetK5` | 5 | $TF_1$-$TF_5$ |
| `fsetK10` | 10 | $TF_1$-$TF_{10}$ |
| `fsetK16` | 16 | $TF_1$-$TF_{16}$ |
| `tsetK1` | 1 | full set (16 tasks) |
| `tsetK3` | 3 | full set (16 tasks) |
| `tsetK7` | 7 | full set (16 tasks) |

### C.2. Protocol (fixed-offset BBOB)

We evaluate all parameter-study settings on the BBOB test suite ($f_1 - f_{24}$) with fixed instance offsets for ease of statistical analysis. We test $d \in \{30, 100, 200, 300, 500\}$ with a budget of $B = 10,000$ function evaluations and population size $\text{pop} = 200$. For each tag, we train 8 independent `MetaDistill` checkpoints (train seeds 0–7) and report mean±std across these checkpoints. We disable test-time fine-tuning, so trends reflect training-data diversity only. Metric definitions are in Section C.1.

### C.3. Diversity tag definitions

Table 7 defines the experimental tags used in our parameter study, specifying the number of training tasks and teachers for each configuration.

### C.4. Full tables

Tables 8 and 9 provide complete numerical results for all parameter-study configurations across all dimensions.

### C.5. Additional visualizations

Figures 21–24 provide visual summaries of the parameter study results, including heatmaps and trend curves for both task diversity and teacher diversity experiments.

### C.6. Best tag summary

Table 10 summarizes the best-performing configuration for each algorithm and dimension, along with the corresponding improvement score and win/loss counts.

*Table 8.* Parameter study: LAD improvement (mean±std over 8 train seeds) on fixed-offset BBOB ($f_1$–$f_{24}$). Higher LAD indicates better `MetaDistill` performance relative to the baseline.

| Tag | $d$=30 | $d$=100 | $d$=200 | $d$=300 | $d$=500 |
|---|---|---|---|---|---|
| **LGA** | | | | | |
| NF_1 | 1.454±0.005 | 1.786±0.007 | 2.084±0.018 | 2.134±0.010 | 2.243±0.008 |
| NF_3 | 1.467±0.003 | 1.800±0.004 | 2.101±0.014 | 2.158±0.014 | 2.256±0.002 |
| NF_5 | 1.487±0.007 | 1.808±0.003 | 2.111±0.015 | 2.180±0.008 | 2.260±0.003 |
| NF_10 | 1.484±0.009 | 1.807±0.003 | 2.121±0.007 | 2.176±0.015 | 2.263±0.004 |
| NF_16 | **1.493±0.006** | **1.813±0.003** | 2.122±0.014 | **2.196±0.015** | **2.268±0.003** |
| NT_1 | 1.467±0.003 | 1.798±0.002 | 2.108±0.013 | 2.152±0.004 | 2.256±0.002 |
| NT_3 | 1.469±0.004 | 1.799±0.003 | 2.113±0.009 | 2.154±0.005 | 2.256±0.002 |
| NT_7 | **1.493±0.006** | **1.813±0.003** | 2.122±0.014 | **2.196±0.015** | **2.268±0.003** |
| **LDE** | | | | | |
| NF_1 | **0.145±0.012** | 0.191±0.013 | 0.174±0.014 | 0.180±0.013 | 0.174±0.014 |
| NF_3 | 0.106±0.183 | 0.333±0.018 | 0.291±0.013 | 0.295±0.013 | 0.287±0.012 |
| NF_5 | −0.123±0.206 | **0.373±0.021** | **0.325±0.012** | **0.325±0.012** | **0.315±0.010** |
| NF_10 | −0.052±0.219 | 0.339±0.010 | 0.298±0.012 | 0.293±0.012 | 0.283±0.012 |
| NF_16 | −0.231±0.027 | 0.347±0.017 | 0.296±0.018 | 0.293±0.020 | 0.282±0.018 |
| NT_1 | −0.258±0.022 | **0.358±0.027** | **0.307±0.024** | **0.302±0.024** | **0.288±0.024** |
| NT_3 | −0.118±0.196 | 0.331±0.022 | 0.283±0.015 | 0.280±0.014 | 0.265±0.015 |
| NT_7 | −0.231±0.027 | 0.347±0.017 | 0.296±0.018 | 0.293±0.020 | 0.282±0.018 |
| **LES** | | | | | |
| NF_1 | −0.490±0.294 | 0.068±0.158 | 0.182±0.093 | 0.221±0.062 | 0.285±0.071 |
| NF_3 | −0.425±0.231 | 0.012±0.312 | 0.136±0.280 | 0.117±0.188 | 0.112±0.228 |
| NF_5 | **0.075±0.184** | 0.149±0.349 | 0.224±0.166 | 0.262±0.109 | 0.254±0.127 |
| NF_10 | −0.344±0.203 | −0.072±0.187 | 0.095±0.158 | 0.140±0.163 | 0.181±0.180 |
| NF_16 | −0.330±0.148 | 0.063±0.239 | 0.240±0.259 | 0.277±0.270 | 0.304±0.284 |
| NT_1 | −0.227±0.360 | **0.226±0.429** | **0.372±0.286** | **0.383±0.258** | **0.399±0.255** |
| NT_3 | −0.328±0.252 | 0.210±0.357 | 0.323±0.212 | 0.356±0.210 | 0.388±0.168 |
| NT_7 | −0.347±0.151 | 0.041±0.223 | 0.233±0.256 | 0.274±0.269 | 0.300±0.282 |
| **POM** | | | | | |
| NF_1 | 2.367±1.629 | 3.359±1.833 | 3.498±1.888 | 3.519±1.903 | 3.582±1.927 |
| NF_3 | **3.517±1.289** | 4.869±1.202 | 5.101±1.248 | 5.188±1.236 | 5.292±1.221 |
| NF_5 | 2.859±1.497 | 4.001±1.602 | 4.061±1.654 | 4.109±1.676 | 4.215±1.709 |
| NF_10 | 2.891±0.635 | 4.175±0.745 | 4.409±0.717 | 4.523±0.731 | 4.570±0.792 |
| NF_16 | 2.630±0.695 | 3.952±0.612 | 4.192±0.580 | 4.275±0.542 | 4.342±0.624 |
| NT_1 | 3.503±0.583 | 4.985±0.590 | 5.191±0.605 | 5.275±0.614 | **5.407±0.633** |
| NT_3 | 3.218±0.693 | **4.993±0.843** | **5.231±0.879** | **5.298±0.875** | 5.398±0.936 |
| NT_7 | 2.221±1.005 | 3.533±1.203 | 3.762±1.264 | 3.767±1.271 | 3.874±1.280 |

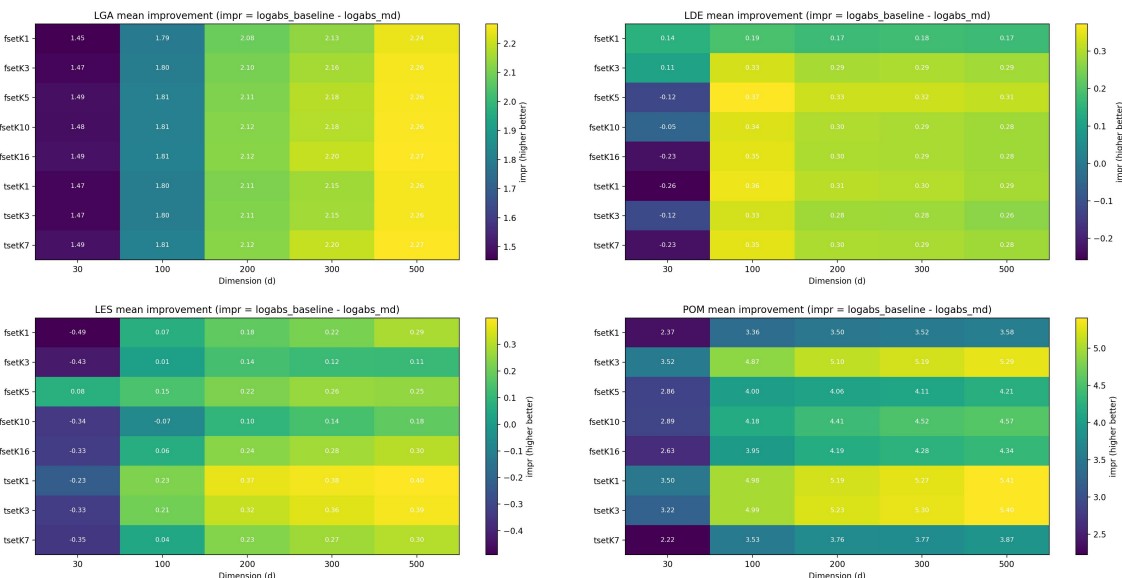

*Figure 21.* Tag–dimension heatmaps for each algorithm. Color encodes LAD (larger is better). We observe consistent gains from increasing the number of training tasks for LGA, while teacher diversity has mixed (algorithm-dependent) effects, with LES generally favoring fewer teachers and POM typically peaking around a moderate teacher set size.

*Table 9.* Parameter study: per-function win/loss counts (mean over 8 train seeds) on fixed-offset BBOB. Each entry shows W / L out of 24 functions (ties are zero throughout and omitted).

| Tag | $d=30$ | $d=100$ | $d=200$ | $d=300$ | $d=500$ |
|---|---|---|---|---|---|
| **LGA** | | | | | |
| NF_1 | 21/3 | 21/3 | 22/2 | 21/3 | 21/3 |
| NF_3 | 21/3 | 21/3 | 22/2 | 21.2/2.8 | 21/3 |
| NF_5 | 21/3 | 21/3 | 22/2 | 22/2 | 21/3 |
| NF_10 | 21/3 | 21/3 | 22/2 | 21.8/2.2 | 21/3 |
| NF_16 | 21/3 | 21/3 | 22/2 | 21.9/2.1 | 21/3 |
| NT_1 | 21/3 | 21/3 | 22/2 | 21/3 | 21/3 |
| NT_3 | 21/3 | 21/3 | 22/2 | 21.2/2.8 | 21/3 |
| NT_7 | 21/3 | 21/3 | 22/2 | 21.9/2.1 | 21/3 |
| **LDE** | | | | | |
| NF_1 | 22.6/1.4 | 22.4/1.6 | 23/1 | 23/1 | 23/1 |
| NF_3 | 22.5/1.5 | 23/1 | 23/1 | 23/1 | 23/1 |
| NF_5 | 22.8/1.2 | 23/1 | 23/1 | 23/1 | 23/1 |
| NF_10 | 22.8/1.2 | 22.9/1.1 | 23/1 | 23/1 | 23/1 |
| NF_16 | 22.6/1.4 | 23/1 | 23/1 | 23/1 | 23/1 |
| NT_1 | 22.8/1.2 | 23/1 | 23/1 | 23/1 | 23/1 |
| NT_3 | 22.8/1.2 | 22.9/1.1 | 23/1 | 23/1 | 23/1 |
| NT_7 | 22.6/1.4 | 23/1 | 23/1 | 23/1 | 23/1 |
| **LES** | | | | | |
| NF_1 | 11.2/12.8 | 14.5/9.5 | 18.5/5.5 | 22.2/1.8 | 22.1/1.9 |
| NF_3 | 11.1/12.9 | 10.9/13.1 | 13/11 | 16.1/7.9 | 15.4/8.6 |
| NF_5 | 15.4/8.6 | 13.4/10.6 | 17.1/6.9 | 19.9/4.1 | 18.1/5.9 |
| NF_10 | 12.6/11.4 | 11.6/12.4 | 13.2/10.8 | 16.5/7.5 | 16.2/7.8 |
| NF_16 | 14.8/9.2 | 15.1/8.9 | 18.2/5.8 | 19.6/4.4 | 19.6/4.4 |
| NT_1 | 14.8/9.2 | 17/7 | 18.5/5.5 | 20.6/3.4 | 20.1/3.9 |
| NT_3 | 15.4/8.6 | 17.1/6.9 | 19.4/4.6 | 21.1/2.9 | 21.9/2.1 |
| NT_7 | 14.4/9.6 | 15/9 | 18/6 | 19.6/4.4 | 19.6/4.4 |
| **POM** | | | | | |
| NF_1–NT_7 | 23/1 | 23/1 | 23/1 | 23/1 | 23/1 |

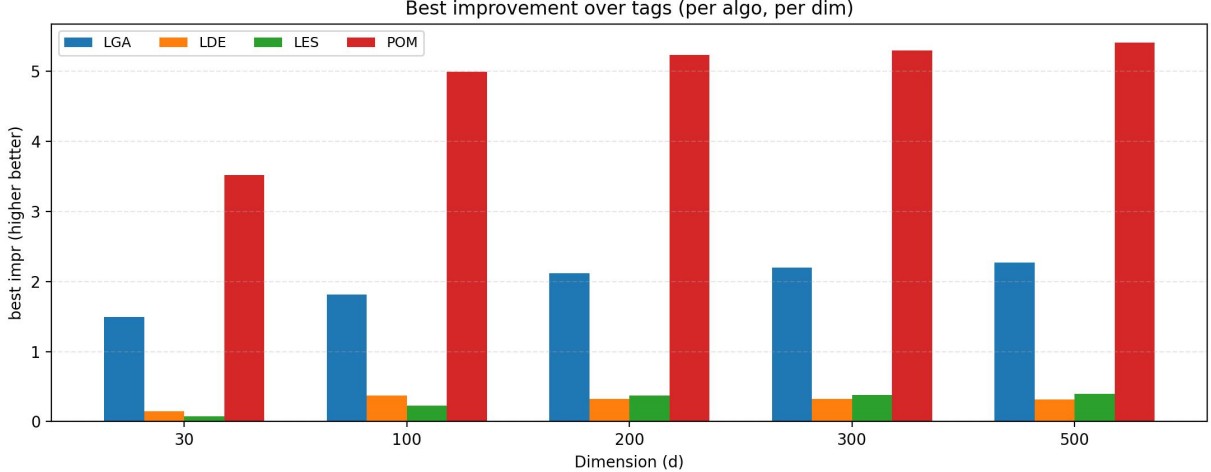

*Figure 22.* Best-tag LAD by algorithm and dimension. Each bar shows the highest mean LAD among all tags for the corresponding (algorithm, dimension).

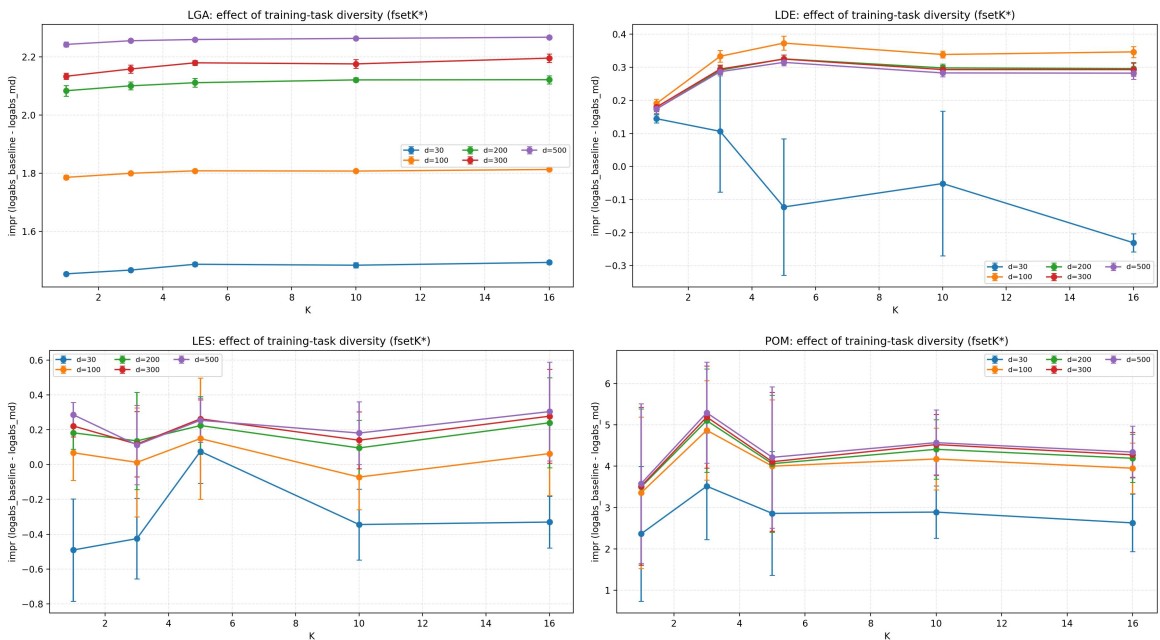

*Figure 23.* Improvement vs. number of training tasks ($K$) for each algorithm (`fsetK*`). Each curve corresponds to a problem dimension.

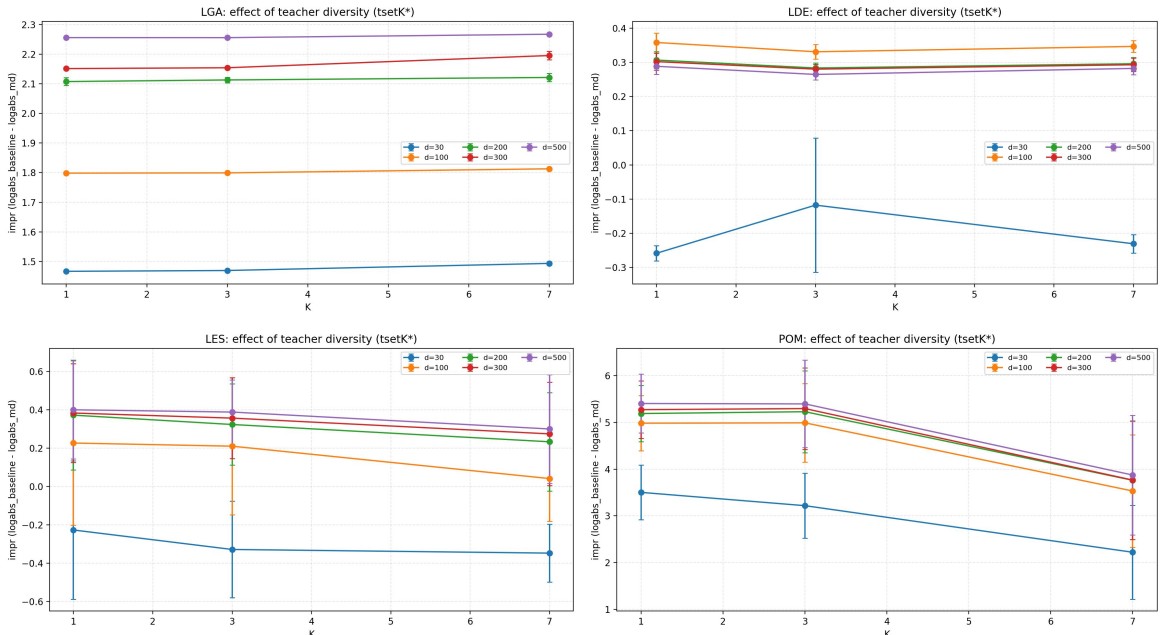

*Figure 24.* Improvement vs. number of teachers ($K$) for each algorithm (`tsetK*`). Each curve corresponds to a problem dimension.

*Table 10.* Best-performing tag per algorithm and dimension (selected by highest mean LAD).

| Algo | Dim | Best Tag | Best LAD (mean±std) | md W/T/L (mean) |
|------|-----|----------|---------------------|-----------------|
| LGA | 30 | fsetK16 | 1.493±0.006 | 21.0/0.0/3.0 |
| LGA | 100 | fsetK16 | 1.813±0.003 | 21.0/0.0/3.0 |
| LGA | 200 | fsetK16 | 2.122±0.014 | 22.0/0.0/2.0 |
| LGA | 300 | fsetK16 | 2.196±0.015 | 21.9/0.0/2.1 |
| LGA | 500 | fsetK16 | 2.268±0.003 | 21.0/0.0/3.0 |
| LDE | 30 | fsetK1 | 0.145±0.012 | 22.6/0.0/1.4 |
| LDE | 100 | fsetK5 | 0.373±0.021 | 23.0/0.0/1.0 |
| LDE | 200 | fsetK5 | 0.325±0.012 | 23.0/0.0/1.0 |
| LDE | 300 | fsetK5 | 0.325±0.012 | 23.0/0.0/1.0 |
| LDE | 500 | fsetK5 | 0.315±0.010 | 23.0/0.0/1.0 |
| LES | 30 | fsetK5 | 0.075±0.184 | 15.4/0.0/8.6 |
| LES | 100 | tsetK1 | 0.226±0.429 | 17.0/0.0/7.0 |
| LES | 200 | tsetK1 | 0.372±0.286 | 18.5/0.0/5.5 |
| LES | 300 | tsetK1 | 0.383±0.258 | 20.6/0.0/3.4 |
| LES | 500 | tsetK1 | 0.399±0.255 | 20.1/0.0/3.9 |
| POM | 30 | fsetK3 | 3.517±1.289 | 23.0/0.0/1.0 |
| POM | 100 | tsetK3 | 4.993±0.843 | 23.0/0.0/1.0 |
| POM | 200 | tsetK3 | 5.231±0.879 | 23.0/0.0/1.0 |
| POM | 300 | tsetK3 | 5.298±0.875 | 23.0/0.0/1.0 |
| POM | 500 | tsetK1 | 5.407±0.633 | 23.0/0.0/1.0 |

*Table 11.* **Robot control task details.** Observation dimension, action space, action mapping, and policy parameter dimension $d$.

| Environment | Obs. dim | Action space | Action mapping | Policy $d$ |
|---|---|---|---|---|
| LunarLander | 8 | Discrete (4) | $\mathrm{argmax}(\mathrm{softmax}(\mathbf{z}))$ | 420 |
| BipedalWalker | 24 | Continuous (4D) | $\tanh(\mathbf{z})$ | 932 |
| Acrobot | 6 | Discrete (3) | $\mathrm{argmax}(\mathrm{softmax}(\mathbf{z}))$ | 323 |

*Table 12.* **Final returns for LES (mean $\pm$ std).** Higher is better. Rows: `baseline` and `md_j{0,1,3,5}`.

| Variant | LunarLander | BipedalWalker | Acrobot |
|---|---|---|---|
| `baseline` | $195.627 \pm 152.976$ | $12.540 \pm 30.804$ | $-64.000 \pm 0.000$ |
| `md_j0` | $280.959 \pm 8.325$ | $29.681 \pm 18.248$ | $-63.667 \pm 0.577$ |
| `md_j1` | $292.529 \pm 1.026$ | $19.552 \pm 14.621$ | $-63.667 \pm 0.577$ |
| `md_j3` | $293.821 \pm 26.896$ | $39.469 \pm 6.377$ | $-64.000 \pm 0.000$ |
| `md_j5` | $286.217 \pm 6.895$ | $29.752 \pm 16.288$ | $-64.000 \pm 0.000$ |

# D. Neuroevolution Setup and Additional Results

## D.1. Robot task setup and evaluation protocol

**Tasks and policy parameterization.** We evaluate three Gymnasium environments (Table 11). Each candidate solution is a policy parameter vector $x \in [-5, 5]^d$ that instantiates a fixed MLP policy with **one hidden layer of width 32** (two linear layers: input→32→output) and leaky ReLU activations. For discrete-action tasks, the network outputs logits and selects the action via $\mathrm{argmax}(\mathrm{softmax}(\mathbf{z}))$. For continuous-action tasks, the network outputs $\tanh$-squashed actions, scaled to the environment action bounds.

**Objective and reported return.** Fitness is defined as **negative episodic return** (a minimization objective):

$$f(x) = -\sum_{t=0}^{T-1} r_t,$$

where $T \leq 256$ is the rollout length (terminated early if the environment ends). We report **return** as $R(x) = -f(x)$, so higher is better.

**Neuroevolution budget.** All methods use the same protocol: population size 200, 500 generations, and seeds $\{0, 1, 2\}$. Each candidate is evaluated with a single rollout of up to 256 environment steps, yielding a maximum of $200 \times 500 = 100{,}000$ candidate evaluations and $200 \times 500 \times 256 = 25.6 \times 10^6$ environment steps per run.

## D.2. Full return results (all tasks, all variants)

Tables 12–15 report final episodic returns (mean $\pm$ std over seeds) for all variants after 500 generations. "Baseline" refers to the underlying neuroevolution optimizer without `MetaDistill`. For readability, we denote `MetaDistill` variants as `md_j0`, `md_j1`, `md_j3`, and `md_j5`. All variants share the same neuroevolution budget described in Appendix D.1.

## D.3. Additional learning curve plots

Figure 25–Figure 27 show mean best-return curves for each task and optimizer, including `baseline` and `md_j{0,1,3,5}`. Figure 28 provides the task-level baseline vs. best-`MetaDistill` comparison for Acrobot. Figure 29 compares LGA vs. LGA+ on Acrobot (see Figure 4 in the main text for LunarLander and BipedalWalker).

*Table 13.* **Final returns for LDE (mean ± std).** Higher is better. Rows: `baseline` and `md_j{0,1,3,5}`.

| Variant | LunarLander | BipedalWalker | Acrobot |
|---------|-------------|---------------|---------|
| `baseline` | $298.954 \pm 7.088$ | $21.098 \pm 5.113$ | $-57.000 \pm 0.000$ |
| `md_j0` | $296.461 \pm 5.446$ | $27.154 \pm 3.119$ | $-56.667 \pm 0.577$ |
| `md_j1` | $299.196 \pm 9.956$ | $26.470 \pm 7.620$ | $-56.333 \pm 0.577$ |
| `md_j3` | $301.686 \pm 12.237$ | $22.136 \pm 4.934$ | $-56.667 \pm 0.577$ |
| `md_j5` | $299.091 \pm 6.384$ | $25.755 \pm 6.633$ | $-56.667 \pm 0.577$ |

*Table 14.* **Final returns for POM (mean ± std).** Higher is better. Rows: `baseline` and `md_j{0,1,3,5}`.

| Variant | LunarLander | BipedalWalker | Acrobot |
|---------|-------------|---------------|---------|
| `baseline` | $286.003 \pm 16.871$ | $9.171 \pm 10.524$ | $-58.667 \pm 3.786$ |
| `md_j0` | $280.630 \pm 10.953$ | $12.534 \pm 3.409$ | $-57.000 \pm 0.000$ |
| `md_j1` | $307.141 \pm 7.822$ | $12.771 \pm 1.012$ | $-57.333 \pm 0.577$ |
| `md_j3` | $283.568 \pm 3.294$ | $11.053 \pm 1.232$ | $-57.000 \pm 0.000$ |
| `md_j5` | $267.022 \pm 10.806$ | $11.321 \pm 1.818$ | $-57.000 \pm 0.000$ |

*Table 15.* **Final returns for LGA vs. LGA+.** Higher is better.

| Version | Variant | LunarLander | BipedalWalker | Acrobot |
|---------|---------|-------------|---------------|---------|
| LGA | `baseline` | $111.915 \pm 161.801$ | $18.401 \pm 12.714$ | $-66.000 \pm 4.359$ |
| | `md_j0` | $-48.767 \pm 82.077$ | $6.735 \pm 16.825$ | $-72.667 \pm 14.154$ |
| | `md_j1` | $-36.268 \pm 63.396$ | $14.579 \pm 18.780$ | $-92.333 \pm 27.025$ |
| | `md_j3` | $-50.522 \pm 131.477$ | $14.594 \pm 16.948$ | $-75.000 \pm 10.536$ |
| | `md_j5` | $-81.340 \pm 3.822$ | $7.652 \pm 11.388$ | $-67.000 \pm 5.196$ |
| LGA+ | `baseline` | $302.710 \pm 5.141$ | $33.495 \pm 6.462$ | $-60.000 \pm 1.000$ |
| | `md_j0` | $312.744 \pm 11.625$ | $24.484 \pm 5.747$ | $-59.333 \pm 1.528$ |
| | `md_j1` | $296.943 \pm 2.444$ | $35.231 \pm 4.018$ | $-60.000 \pm 1.732$ |
| | `md_j3` | $316.111 \pm 9.534$ | $41.406 \pm 6.406$ | $-61.667 \pm 2.309$ |
| | `md_j5` | $304.783 \pm 7.514$ | $37.914 \pm 3.884$ | $-59.000 \pm 0.000$ |

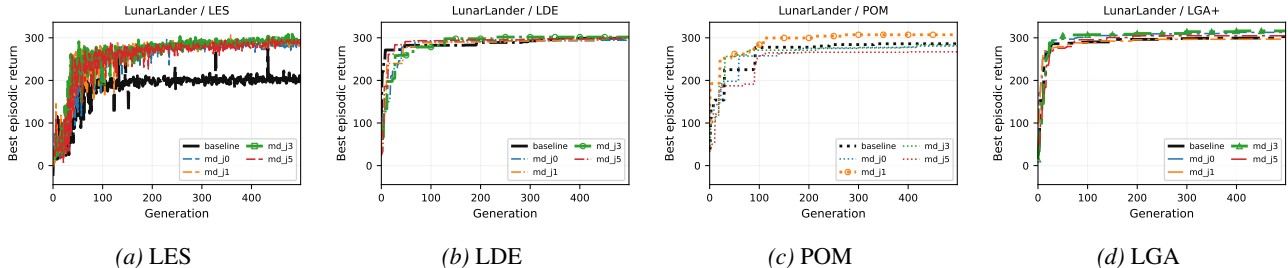

*(a)* LES  *(b)* LDE  *(c)* POM  *(d)* LGA

*Figure 25.* LunarLander: mean best episodic return vs. generation for `baseline` and `md_j{0,1,3,5}` (averaged over seeds).

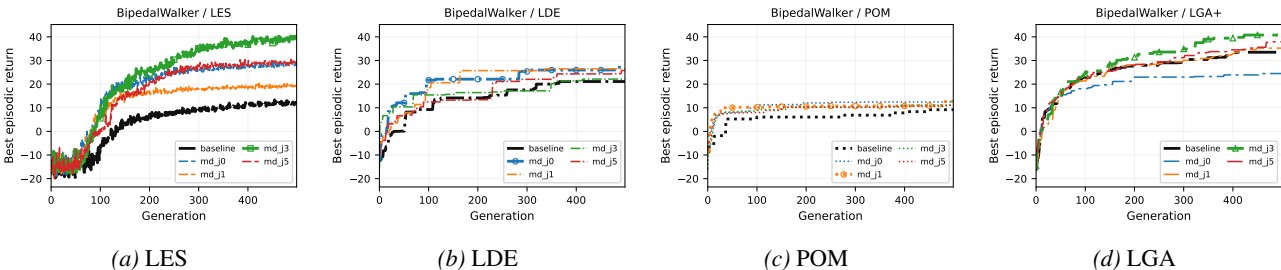

*(a)* LES  *(b)* LDE  *(c)* POM  *(d)* LGA

*Figure 26.* BipedalWalker: mean best episodic return vs. generation for `baseline` and `md_j{0,1,3,5}` (averaged over seeds).

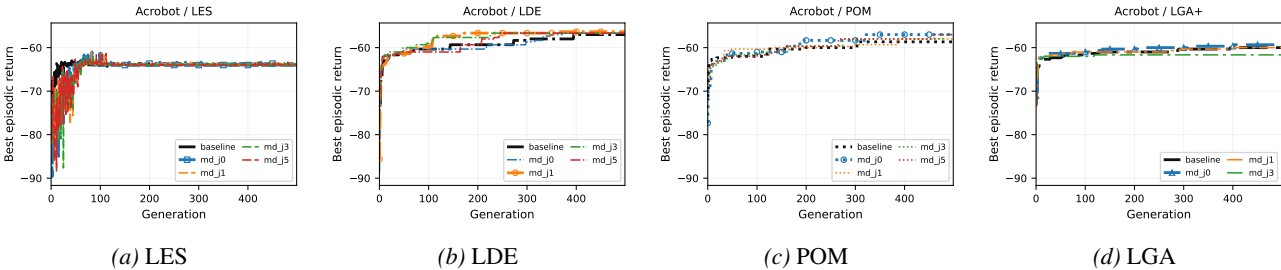

*(a)* LES      *(b)* LDE      *(c)* POM      *(d)* LGA

*Figure 27.* Acrobot: mean best episodic return vs. generation for `baseline` and `md_j{0,1,3,5}` (averaged over seeds).

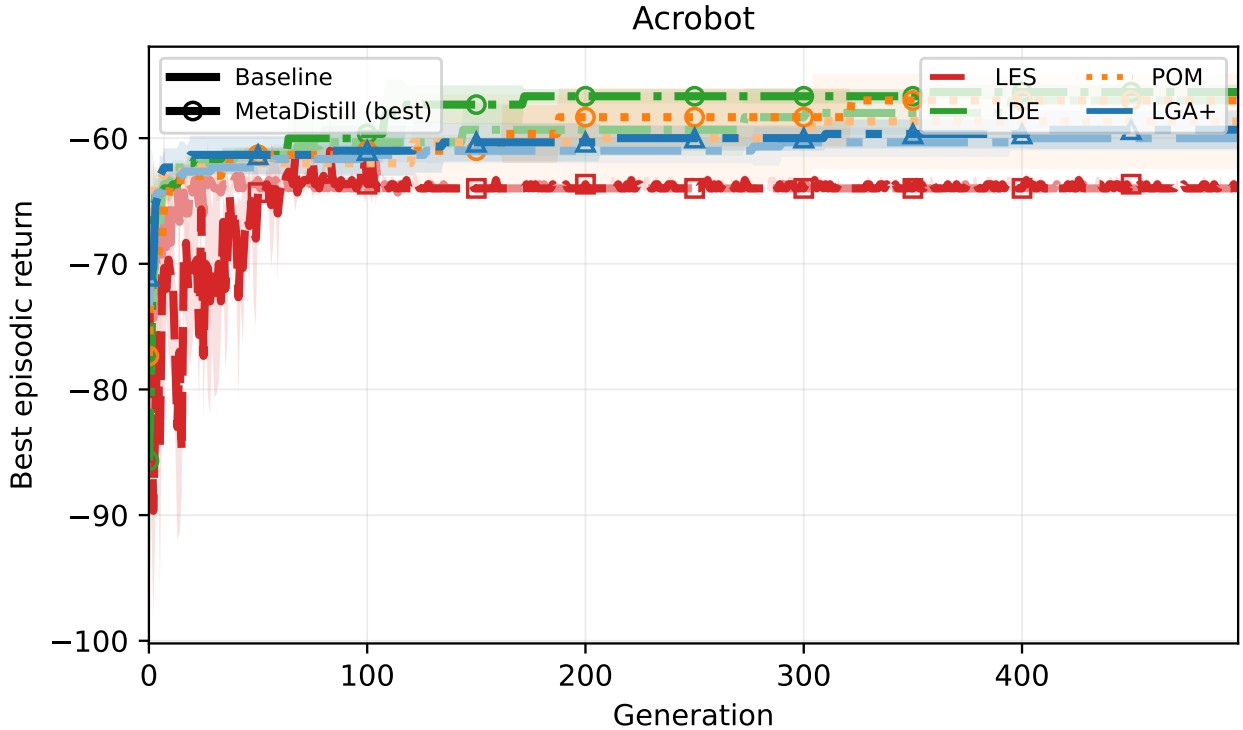

*Figure 28.* Acrobot: task-level comparison of baseline vs. best `MetaDistill` variant (selected by highest mean return). Lines show mean over seeds; shaded areas indicate ±1 std.

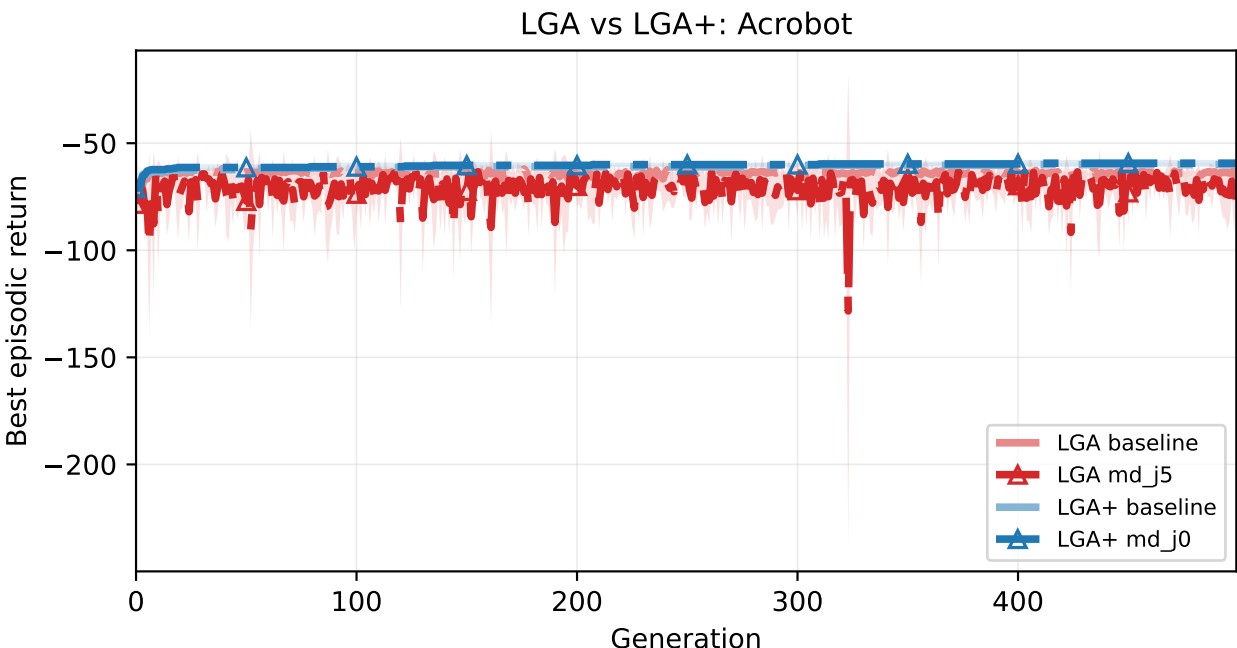

*Figure 29.* LGA vs. LGA+ on Acrobot. Baseline and best `MetaDistill` variant are selected among `md_j{0,1,3,5}` by highest mean return. Lines show mean over available seeds; shaded areas indicate $\pm 1$ std.

