# OpenReview forum: "$\texttt{MetaDistill}$: Unlocking the Performance Ceiling for Pretrained Optimizers"
_ICML.cc/2026/Conference — ICML 2026 regular_

### Official Review · Reviewer_qt7i · 2026-03-07

**Soundness:** 3
**Presentation:** 3
**Significance:** 3
**Originality:** 3
**Overall Recommendation:** 5
**Confidence:** 5

**Summary:**

This paper proposes a general MetaBBO training framework to train the meta-level agent of MetaBBO by distilling the optimization behaviours of advanced BBO methods. The authors collect a optimization trajectory set by running a set of BBO optimizers on the target problem set and assemblethe effective optimization intervals. A distillation mechanism is designed the train the MetaBBO policy from the collected trajectories and a test-time self-supervised fine-tuning is introduced to further enhance the performance. Experimental reults demonstrae the effectiveness of the proposed framework on some MetaBBO methods.

**Compliance With Llm Reviewing Policy:**

Affirmed.

**Final Justification:**

The detailed response addresses my concerns. The additional experimental results validate the effectiveness of the proposed method. So I raise the score.

**Key Questions For Authors:**

1. What is the definition of "most effective" in assembling a refined optimization trajectory? Could this lead to a fragmented trajectory or a greedy strategy?

2. The population size settings for the baselines in the experiments differ from those used in their original papers. Since the training is population size agnostic, how does the performance of policies trained by MetaDistill compare to that of the original policies under their original population sizes?

3. How does the performance of policies trained by MetaDistill compare to that of the teacher models?

4. What is the meaning of "teacher-synth" in Figs. 5–20?

**Limitations:**

yes

**Strengths And Weaknesses:**

Strengths:
1. Using advanced BBO methods as teachers and learning from their optimization strategies allows the proposed framework to leverage decades of optimization experience to enhance MetaBBO performance. The distribution-centric approach enhances generalizability and enables MetaBBO to learn from a variety of BBO methods.

2. The introduced test-time self-supervised fine-tuning adapts the MetaBBO policy to new problems and further boosts optimization performance. Using a population-based loss instead of objective values extends the applicability of the framework.

3. Experimental results show that the framework successfully improves the performance of several MetaBBO methods, enabling them to outperform their original versions on both synthetic problems and neuroevolution tasks.

Weaknesses:
1. Training the proposed framework requires running a set of BBO methods on the target problem set, which may consume substantial computational resources and time.

2. While using the mean and covariance matrix of populations allows the framework to incorporate more BBO teachers, for BBO methods that sample populations uniformly, such as DE and PSO, the mean and covariance matrix may not adequately represent the population distribution.

3. Applying the framework to different MetaBBO methods and optimization problems may require different hyperparameters (e.g., the SSFT interval), which raises the barrier to entry for users.

---

> ### Author Rebuttal · Authors · 2026-03-31
>
> We appreciate the reviewer for all the comments.
> > W1 Training the ...
>
> We agree that collecting offline trajectories increases the computational burden, but it is not significant, but controllable and amortized. First, it can be flexibly adjusted via trajectory length, number of teachers and training tasks, and population size of teachers. Second, trajectory collection is a one-time preprocessing step: new data is only needed when new teachers or tasks are introduced, and can be reused across multiple training runs. This is similar in spirit to standard offline dataset construction in many learning paradigms. Importantly, excluding trajectory collection, the training cost of MetaDistill is comparable to the original framework, and even when including it, the performance gains consistently justify this trade-off in our experiments.
> | framework (training POM with 128 epochs)                     | GPU hours (RTX 5090) |
> | ------------------------------------------------------------ | -------------------- |
> | original                                                     | 0.1600               |
> | MetaDistill (w/o constructing knowledge base)                | 0.2195               |
> | constructing knowledge base (80 trajectories, the same size as the one used for training in the paper) | 0.0378               |
> > W2 While using ...
>
> While DE and PSO often uniformly sample initial population, this property only holds at the first iteration. As optimization proceeds, their population distributions are rapidly altered by mutation, recombination, velocity updates, and selection mechanisms. In particular, DE introduces bias through differential mutation and selection, while PSO drives particles toward personal and global best positions, resulting in strong aggregation effects. More fundamentally, any optimization algorithm with selection pressure necessarily induces non-uniformity in the population distribution; otherwise, it would reduce to random search. Therefore, the populations used during distillation are not uniformly distributed in practice, but instead exhibit concentration and structural patterns. In this context, representing populations using mean and covariance provides a compact and effective approximation of their distribution, enabling scalable distillation across heterogeneous BBO optimizers. While this approximation may not capture all higher-order characteristics, it strikes a practical balance between expressiveness and computational efficiency.
> > W3 Applying the framework ...
>
> MetaDistill is relatively insensitive to hyperparameters. Taking the SSFT interval j as an example, The following table reports the best performance achieved by the MetaDistill variants of LES under different SSFT intervals. These results show that a significant performance improvement only occurs when the SSFT interval is adjusted from j=0 (no SSFT) to j!=1, while there is little difference among non-zero intervals.
> |j|improvement over baseline|
> |---|---:|
> |0|0.824990|
> |1|0.872302|
> |best|0.872304|
> > Q1 What is the definition ...
>
> "Most effective" means relatively best fitness value among all teachers. We agree that an excessively small distillation interval $i$ can lead to fragmented trajectories and frequently changing distillation targets, resulting in greedy stragety and unstable training. Therefore, in our experiments, we set $i=10$, an empirically optimal value that balances exploitation and exploration while promoting training stability.
> > Q2 The population size ...
>
> We compare the peformance of MetaDistill variants against their baselines in their original population size setting on BBOB (d=100) under evaluation budget B=10000, the results show that the improvement of learnable optimizers by MetaDistill is independent of population size.
> |optimizer|population size (same as original paper)|improvement over baselines|
> |---|---|---|
> |LES|16| 0.4303|
> |LGA|32| 0.2661|
> |POM|100| 3.6956|
> |LDE|50| 0.2623|
> > Q3 How does the performance ...
>
> Figure 5-20 in the paper already provides Teacher-Synth curves as a teacher-level reference, where "Teacher-Synth" indicates the construction of the optimal trajectory using all teachers' trajectories collected on each test task in the same way as the training trajectories, representing the optimal level of all teachers. These figures shows that our MetaDistill variants are comparable or even surpass their teachers in most cases, in brief, our MetaDistill variants are superior to or tie to teacher optimizers in 10/16 scenarios (4 LOs * 4 BBOB dimensional settings).
> > Q4 What is the meaning ...
>
> Please see our response to **Q3**.

---

> > ### Author Rebuttal · Reviewer_qt7i · 2026-04-04
> >
> > The response resolves my questions.

---

> > > ### Author Response · Authors · 2026-04-05
> > >
> > > **Dear Reviewer**,
> > >
> > > Thank you for your support and valuable feedback throughout the review process. We are grateful for the time and expertise you dedicated to improving our paper. Your insights have undoubtedly strengthened our work.
> > >
> > > Rest assured, we will carefully consider all points discussed during this review cycle as we work to further refine and improve the paper.
> > >
> > > **Sincerely,**
> > >
> > > **Authors**

---

### Official Review · Reviewer_jxtx · 2026-03-09

**Soundness:** 3
**Presentation:** 3
**Significance:** 3
**Originality:** 2
**Overall Recommendation:** 4
**Confidence:** 3

**Summary:**

Most existing MetaBBO methods rely on learning-from-scratch via self-supervised or unsupervised objectives, which restricts the meta-optimizer to suboptimal strategies due to a lack of exposure to high-quality optimization trajectories.To bridge this gap, the authors introduce MetaDistill, a universal training framework designed to internalize expert knowledge from classical heuristics. The framework consists of: Diversity-Preserving Distillation (DPD), Self-Supervised Fine-Tuning (SSFT).  The study investigates the impact of task and teacher diversity on generalization, noting that excessive diversity can reach a saturation point or even degrade performance. Experimental results across BBOB benchmarks and reinforcement learning tasks demonstrate the framework's effectiveness in enhancing the convergence and solution quality of various learnable optimizers.

**Compliance With Llm Reviewing Policy:**

Affirmed.

**Final Justification:**

The authors’ rebuttal addressed my concerns.

**Key Questions For Authors:**

### Robustness to Suboptimal Experts

The proposed framework relies heavily on offline trajectories generated by expert optimizers. This raises an important concern: if the selected experts are suboptimal on certain landscape types, the student may inherit biased or ineffective search behaviors. The paper would be strengthened by clarifying how this issue is mitigated. In particular, is there any mechanism to evaluate, filter, or reweight expert trajectories based on their task-specific performance, either during dataset construction or throughout the distillation process?

### Computational Budget and Efficiency

Although MetaDistill is motivated by improving optimization efficiency, its overall computational cost is not yet sufficiently clear. The DPD stage requires large-scale expert trajectory collection and pre-training, which may introduce substantial overhead. It would therefore be helpful to provide a quantitative comparison of the total cost of MetaDistill, including data collection and training time, against learning-from-scratch baselines. In addition, for the SSFT stage, please clarify how many additional function evaluations are typically required before the student achieves a noticeable performance gain.

### Sensitivity to the Scaling Factors

The introduction of the scaling factors $s_1$ and $s_2$ for stabilizing KL-divergence minimization is reasonable, but their influence on final performance remains unclear. Please provide more discussion on the sensitivity of the method to these hyper-parameters. Specifically, how were $s_1$ and $s_2$ selected, how robust is the method to different values, and do these factors need to be re-tuned when changing the student architecture or problem dimension?

**Limitations:**

yes

**Strengths And Weaknesses:**

## Strengths
### Novel Perspective on Performance Bottlenecks
The paper provides a compelling motivation by identifying the "performance ceiling" in current MetaBBO research. It correctly points out that learning-from-scratch often traps meta-optimizers in suboptimal strategies, a critical insight for the field.

### Architecture-Agnostic Framework
The introduction of a "population-based distribution" as a universal medium is a highly clever design. This allows for seamless knowledge distillation from diverse, heterogeneous expert algorithms to any learnable student optimizer, regardless of their internal architectures.

## Weaknesses

### Increased Computational Overhead

The reliance on offline trajectory collection and a dedicated pre-training distillation phase adds significant complexity and computational cost to the training pipeline compared to end-to-end methods.

### Dependency on Expert Quality

The performance of MetaDistill is inherently bounded by the quality of the selected "expert" algorithms. The paper lacks a detailed discussion on whether the student might inherit the specific failure modes or biases of the teachers in extremely rare or complex landscapes.

---

> ### Author Rebuttal · Authors · 2026-03-31
>
> We appreciate the reviewer for all the comments.
> > W1 Increased Computational Overhead
>
> We agree that collecting offline trajectories increases the computational burden, but it is not significant, but controllable and amortized. First, it can be flexibly adjusted via trajectory length, number of teachers and training tasks, and population size of teachers. Second, trajectory collection is a one-time preprocessing step: new data is only needed when new teachers or tasks are introduced, and can be reused across multiple training runs. This is similar in spirit to standard offline dataset construction in many learning paradigms. Importantly, excluding trajectory collection, the training cost of MetaDistill is comparable to the original framework, and even when including it, the performance gains consistently justify this trade-off in our experiments.
>
> | framework (training POM with 128 epochs)                     | GPU hours (RTX 5090) |
> | ------------------------------------------------------------ | -------------------- |
> | original                                                     | 0.1600               |
> | MetaDistill (w/o constructing knowledge base)                | 0.2195               |
> | constructing knowledge base (80 trajectories, the same size as the one used for training in the paper) | 0.0378               |
>
> > W2 Dependency on Expert Quality
>
> We agree that the quality of teacher trajectories affects distillation because they determine the quality of the supervision signal. However, our MetaDistill framework mitigates student's inheritance of teachers' failure modes and ensures its robustness to suboptimal teachers in three aspects: (1) On the Construction of the Training Set. Considering that complex landscapes can be viewed as combinations and superpositions of simple landscapes, and the latter are often more general, we carefully selected simple, low-dimensional tasks to construct the training set in order to avoid hindering students' training due to the poor performance of the teacher optimizer on the training tasks. (2) On the Filtering of trajectories.  As described in Section 3.2 of our paper, we perform fitness-based filtering while constructing the trajectories. Specifically, we divide each teacher's optimization trajectory on the same task into segments of length i (i.e., distillation intervals), and select the segment with the best final fitness value among all teachers to construct the teacher trajectory for distillation. For example, if i=2 and the maximum generation T=5, for the task $\min f(x)$, teacher A has a trajectory $\tau_A = [(x_{A1}, 10), (x_{A2}, 6), (x_{A3}, 3), (x_{A4}, 2),(x_{A5}, 1)]$; teacher B has a trajectory $\tau_B = [(x_{B1}, 9), (x_{B2}, 8), (x_{B3}, 4), (x_{B4}, 3),(x_{B5}, 0)]$, then the final optimal trajectory used for distillation is $\tau = [x_{A1}, x_{A2}, x_{A3}, x_{A4}, x_{B5}]$. Through this method, we can filter out the suboptimal segments from all teachers.  (3) On Test Time Fine-Tuning.  The distilled optimizer's optimization strategy is not frozen. Instead, SSFT can fine-tune the strategy based on the downstream target task, thereby preventing students from carrying over the learned suboptimal strategy, which we believe is unlikely to happen due to (1) and (2), to the target task.
>
> > Q1 Robustness to Suboptimal Experts
>
> Please see our response to **W2**.
>
> > Q2 Computational Budget and Efficiency
>
> To clarify, we do not need additional function evaluations while performing SSFT, please refer to Fig. 5-20 of our paper for details, all optimizers used the same evaluation budget and population size settings. During SSFT, we only need the evaluated fucntion value or population generated by the distilled optimizer to tune its parameters using Eq. 5 or 6 in our paper. Regarding the computional cost, please see our response to **W1**.
>
> > Q3 Sensitivity to the Scaling Factors
>
> To clarify, $s_1$, $s_2^{\mu}$, and $s_2^{\Sigma}$ are not tuned hyperparameters. According to Eq. 4 of our paper, $s_1$ is the reciprocal of the square root of the dimension $d$, so it depends only on problem dimension. $s_2^{\mu}$ and $s_2^{\Sigma}$ are the reciprocals of the largest absolute elements in the mean and covariance terms, respectively, so they are determined by the input distribution statistics of $\mathcal{L}_1$ rather than manually tuned. Our ablation below is conducted on LES on BBOB. For their influence on final performance, the ablation study shows that these normalization terms play a crucial role in training stability; w/o normalization of the KL divergence, the performance of the distilled optimizer degrades.
>
> | setting     | improvement over baseline |
> | ----------- | ------------------------: |
> | MetaDistill |                    0.4031 |
> | w/o KL norm |                    0.1196 |

---

> > ### Author Rebuttal · Reviewer_jxtx · 2026-04-03
> >
> > Thank you for your detailed response. I believe your research is meaningful, and I appreciate your clarification of my questions.

---

> > > ### Author Response · Authors · 2026-04-05
> > >
> > > **Dear Reviewer**,
> > >
> > > Thank you for your support and valuable feedback throughout the review process. We are grateful for the time and expertise you dedicated to improving our paper. Your insights have undoubtedly strengthened our work.
> > >
> > > Rest assured, we will carefully consider all points discussed during this review cycle as we work to further refine and improve the paper.
> > >
> > > **Sincerely,**
> > >
> > > **Authors**

---

### Official Review · Reviewer_jFG1 · 2026-03-10

**Soundness:** 4
**Presentation:** 3
**Significance:** 3
**Originality:** 4
**Overall Recommendation:** 5
**Confidence:** 4

**Summary:**

This paper proposes MetaDistill, a framework that distills trajectories from classical BBO algorithms (CMA-ES, JADE, PSO, GA, etc.) into learnable black-box optimizers via population-level distributional matching using forward KL divergence. The approach is evaluated in the context of meta-training four learnable optimizers (LES, LGA, LDE, POM) across BBOB and three control tasks. The results show strong and consistent performance improvements from using MetaDistill across all tasks.

**Compliance With Llm Reviewing Policy:**

Affirmed.

**Final Justification:**

The authors have resolved my soundness concerns about untuned baselines and cherry-picked comparisons in Table 1. I no longer have concerns about the paper and I recommend accepting it.

**Key Questions For Authors:**

1. How does the best MetaDistill variant for each learned optimizer compare against the teacher algorithms (e.g., CMA-ES, L-SHADE, etc.) on BBOB with a similar tuning budget to meta-training?
2. Why does MetaDistill produce negative transfer for LES in 6/8 configurations at d=30?
3. How extensive was the tuning of teacher methods? Would it be possible to unlock better performance with more tuning?

**Limitations:**

Yes

**Strengths And Weaknesses:**

**Strengths**
- The paper is well structured and easy to follow.
- The idea of distilling from multiple teachers to explore a diversity of modes is compelling.
- The authors' experimental study covers a diversity of tasks, including the BBoB suite and various control tasks, showing strong performance of the proposed method in most cases.
- To the best of my knowledge, the proposed method is novel and has not been proposed before within the context of MetaBBO.


**Weaknesses**
- My primary concern is that the paper never compares the distilled optimizers against the teacher algorithms themselves. I see some comparison in the appendix, but I am unsure if they have been sufficiently tuned to be representative, and I do see the teacher methods outperforming in some cases.
- My second concern is that the results are far more uneven than the main text suggests. Table 1 reports cherry-picked best variants by reporting the performance of the best optimizer out of the 5 that were trained for a given task. This reports the best possible performance attainable but requires access to the test set for evaluation, which is not the case in practice. This should be made clear in the caption.
- While using a normalized distribution is necessary, the Gaussian assumption is not thoroughly motivated or examined. For instance, it may not be representative of all problems. Could the authors comment on this further in the paper?
- Missing references. A related but distinct series of works focuses on learning optimizers for supervised tasks [1-7]. It would be interesting to discuss how meta-training methods differ here and how MetaBBO methods could be borrowed for supervised learned optimization and vice-versa.

**Additional comments**
- One interesting application of the method may be to optimize learned optimizers themselves. For instance, a line of related works on learned optimization for supervised tasks [1-7] exclusively use ES to optimize a smoothed version of the meta-loss. Perhaps Meta-Distill could be successfully used here.

---
**References**

[1] [Understanding and correcting pathologies in the training of learned optimizers]

[2] [A Closer Look at Learned Optimization: Stability, Robustness, and Inductive Biases]

[3]  [Learning to Generalize Provably in Learning to Optimize]

[4] [μLO: Compute-Efficient Meta-Generalization of Learned Optimizers]

[5] [Practical Tradeoffs Between Memory, Compute, and Performance in Learned Optimizers, CoLLAs 2022]

[6] [VeLO: Training Versatile Learned Optimizers by Scaling Up]

[7][Meta-learning Optimizers for Communication-Efficient Learning]

---

> ### Author Rebuttal · Authors · 2026-03-31
>
> We appreciate the reviewer for all the comments.
> > W1 My primary ...
>
> First, we have compared the distilled optimizers against the teacher optimizers in Appendix B.2 of our paper. All optimizers used the same evaluation budget and population size settings, and the final results were obtained through multiple independent runs and hyperparameter tuning. Therefore, we indicate that these results are sufficiently representative. Second, "Teacher-Synth" in Figure 5-20 indicates the filtering and synthesis of all teachers' trajectory segments of test tasks in the same way with training trajectories, representing the optimal level of all teachers. Finally, by the No Free Lunch theorem, we cannot expect MetaDistill to outperform the teacher optimizer on all tasks, however, our MetaDistill variants are superior to or tie to teacher optimizers in 10/16 scenarios (4 LOs * 4 BBOB dimensional settings).
>
> > W2 My second ...
>
>  In practice, due to evaluation budget constraints, we are indeed unable to perform multiple optimizations using different parameter settings and then select the best result for some tasks. Actually, we have indicated in the caption of Table 1 that the best variant is selected across different SSFT intervals, but we would like to emphasize this point in the revision. Additionally, the performance gaps of the same MetaDistill variant under different SSFT intervals are far from enough to cause the reviewer's concern. Let's take LES as an example. The following table reports the best performance achieved by the MetaDistill variants of LES under different SSFT intervals. These results show that a significant performance improvement only occurs when the SSFT interval is adjusted from j=0 (no SSFT) to j!=1, while there is little difference among non-zero intervals.
> |j|improvement over baseline|
> |---|---:|
> |0|0.824990|
> |1|0.872302|
> |best|0.872304|
>
> > W3 While using ...
>
> We agree that the Gaussian distribution cannot represent all problems, especially for multimodal problems, which may lead to inevitable distribution mismatch. However, the Gaussian representation serves as an information bottleneck that filters out instance-specific or optimizer-specific details, and instead preserves the dominant search geometry (i.e., search center and principal directions), which is more transferable across tasks.
>
> > W4 Missing references ...
>
> We appreciate the reviewer's suggestion, we will discuss these works in the Related Work section of our revision. Due to character limitations, we have presented only a part of the discussion here:
> **Learn to Optimize** Although MetaBBO and L2O focus on the meta-training of learnable optimizers (LOs) in the fields of black-box optimization and gradient-based optimization, respectively, they are intertwined with each other. For example, many black-box LOs can be pretrained on differentiable problems using first-order LOs, while VeLO[6] leverage an LSTM and MLP as gradient decent operator to learn an update rule, whose parameters are updated via gradients estimated by Evolution Strategies on a diversity of tasks. Besides, Thérien et al.[4] apply $\mu$-parameterization to LOs, primarily to ensure stable width scaling. Interestingly, they find that this approach also significantly enhances meta-generalization to unseen depths and longer training horizons ...{omitted due to character limitations}
>
> > Ad. Co. One interesting ...
>
> Both self-distillation and self-optimization can make fuller use of the distilled optimizer. Specifically, the former can provide a higher-quality distillation source where teacher trajectories are constructed via pretrained MetaDistill variants. While the latter can be used to train other learnable optimizers (e.g. to train LES/LGA themselves or VeLO), or even broader scenarios like LLM fine-tuning. We appreciate the reviewer's valuable comment again, we would like to include these potential applications in our future work.
>
> > Q1 How does ...
>
> Please see our response to **W1**, we have performed such experiments in Appendix B.2 of our paper.
>
> > Q2 Why does ...
>
> As shown in Fig 23-24, LES tends to perform best with low teacher diversity and moderate training task diversity, and the gain of MetaDistill on LES increases with increasing dimensionality d. However, the NF_x experiments in Table 7 were all conducted with NT=7, while the NT_x experiments were all conducted with NF=16, which is not optimal for LES. Therefore, at d=30, MetaDistill brought negative transitions and a small positive transition to LES. Nevertheless, with increasing d, the gain of MetaDistill on LES significantly increases.
>
> > Q3 How extensive ...
>
> To construct the distillation knowledge base, we performed 16 independent optimizations on each training task using each teacher optimizer. We agree that finer hyperparameter searches and larger optimization budgets may yield better distillation results, but this also implies a greater computational burden.

---

> > ### Author Rebuttal · Reviewer_jFG1 · 2026-04-03
> >
> > My concerns have been addressed, so I am raising my score accordingly.
> >
> > Regarding W2, it would be good for the authors to make this clear in the caption.
> >
> > Regarding Q3, it would be helpful to discuss this somewhere in the paper.

---

> > > ### Author Response · Authors · 2026-04-05
> > >
> > > **Dear Reviewer**,
> > >
> > > Thank you for your support and valuable feedback throughout the review process. We sincerely appreciate your decision to raise the score—this means a great deal to us. We are grateful for the time and expertise you dedicated to improving our paper. Your insights have undoubtedly strengthened our work.
> > >
> > > Rest assured, we will carefully consider all points discussed during this review cycle as we work to further refine and improve the paper.
> > >
> > > **Sincerely,**
> > >
> > > **Authors**

---

### Official Review · Reviewer_d5Dn · 2026-03-13

**Soundness:** 3
**Presentation:** 2
**Significance:** 2
**Originality:** 3
**Overall Recommendation:** 4
**Confidence:** 2

**Summary:**

This paper proposes a broader training framework. It first collects trajectories from several classical optimizers. Then, converts them into a common population-distribution representation, and trains the student optimizer using diversity-preserving distillation.
Finally, they optionally refine it at test time with SSFT. The paper’s contribution is not a new low-level optimizer architecture, but a training paradigm for pretrained optimizers.

**Compliance With Llm Reviewing Policy:**

Affirmed.

**Final Justification:**

The authors have clarified their considerations on Gaussian-representation in both heuristic and quantitative ways. I thereby decided to increase my score.

**Key Questions For Authors:**

1. It seems like the method depends on an offline trajectory knowledge base, which is a big practical cost. To use MetaDistill, one may first need to generate a reasonably rich and high-quality database of expert trajectories. This adds preprocessing cost and design complexity. Could you please discuss more on this issue?
2. Similar in the weakness part, the distribution mismatch may result in mis-capture of algorithm-specific structure. The shared representation is a strength, but also a limitation. Some optimizer-specific signals may be lost when everything is compressed into $(\mu, \Sigma)$. Could you explain more about the distribution mismatch problem?
3. The framework is broad, but ablations appear limited given that the full system includes: trajectory filtering, multiple teachers, etc. Is it possible to conduct deeper ablations on which component contributes most?

**Limitations:**

yes

**Strengths And Weaknesses:**

Strengths:
1. One of the good points the paper’s framing is that it identifies a real and specific weakness in existing MetaBBO: trial-and-error meta-training may indeed produce strong but bounded strategies, especially when the optimizer never observes high-quality exemplars. That is a sensible motivation.
2. The paper tests on the BBOB with dimensions 30, 100, 200, and 300. This is a good generalization setting because training is apparently done on much smaller 10D synthetic functions.
That means the generalization claim is not trivial.

Weaknesses:
1. The paper tends to present the field in a somewhat rigid criterion, something like: self-supervised = ceiling-limited, distillation from experts=higher ceiling. That is to some extend plausible, but slightly simplified. In reality, the performance gap may come from several problem-dependent factors, including but not limited to stronger supervision, better initialization, richer training data. So the historical narrative is coherent, but a bit beautified toward the paper’s main idea.
2. There is no formal proof of the ''performance ceiling'' claim. The paper’s central conceptual claim is that scratch-based training has a restricted upperbound and expert distillation lifts the bound. But this is not proven formally. What the paper actually gives is a plausible learning argument and empirical evidence that distilled variants do better, which is supportive, but not a rigorous theorem.
3. Representing a population only by mean and covariance is convenient, but it may discard important multimodal or structured behavior. The paper does not rigorously analyze what is lost by this compression resulted by population Gaussian summary.

---

> ### Author Rebuttal · Authors · 2026-03-31
>
> We appreciate the reviewer for all the comments.
> > W1 The paper tends to ...
>
> We clarify that we do not claim a strict dichotomy such as “self-supervised=performance-ceiling limited” and “distillation= performance-ceiling lifted,” nor do we claim a formal upper-bound theorem. Instead, our “performance ceiling” refers to an empirical limitation observed in learning-from-scratch paradigms, where learnable optimizers (LOs) must iteratively improve using their own historically collected, often suboptimal data, forming a self-referential loop that makes it difficult to align with higher-quality optimization strategies. Therefore, we treat the performance ceiling as an empirically grounded phenomenon rather than a formal bound, and MetaDistill as one effective (but not exclusive) way to mitigate any potential performance ceiling limitations. We also agree that multiple factors can influence the performance of LOs. Here, we evaluate the improvements to LES achieved by two supervised learning approaches: MetaDistill and global optima guidance on BBOB. The results show that, although global optima guidance leads to performance gains over self-supervised learning, it still underperforms MetaDistill, despite the potentially higher computational cost required to obtain global optima.
>
> | setting                 | improvement over SS-only |
> | ----------------------- | -----------------------: |
> | MetaDistill (KL + norm) |                 +11.2363 |
> | global optima guidance  |                 +10.7633 |
> > W2 There is no formal ...
>
> Please see our response to **W1**.
>
> > W3 Representing a population ...
>
> We agree that distribution mismatch is inevitable. Representing a population with a Gaussian $(\mu, \Sigma)$ compresses the teacher distribution to first- and second-order statistics, which may discard multimodality and optimizer-specific strategies when the true population is non-Gaussian. However, this compression is intentional. The Gaussian representation serves as an information bottleneck that filters out instance-specific or optimizer-specific details, and instead preserves the dominant search geometry (i.e., search center and principal directions), which is more transferable across tasks.
>
> > Q1 It seems like ...
>
> We agree that MetaDistill requires collecting expert trajectories, which introduces additional offline cost. However, this cost is both controllable and amortized. First, it can be flexibly adjusted via trajectory length, number of teachers and training tasks, and population size of teachers. Second, trajectory collection is a one-time preprocessing step: new data is only needed when new teachers or tasks are introduced, and can be reused across multiple training runs. This is similar in spirit to standard offline dataset construction in many learning paradigms. Importantly, excluding trajectory collection, the training cost of MetaDistill is comparable to the original framework, and even when including it, the performance gains justify this trade-off in our experiments.
>
> | framework (training POM with 128 epochs)                     | GPU hours (RTX 5090) |
> | :----------------------------------------------------------- | -------------------- |
> | original                                                     | 0.1600               |
> | MetaDistill (w/o constructing knowledge base)                | 0.2195               |
> | constructing knowledge base (80 trajectories, the same size as the one used for training in the paper) | 0.0378               |
>
> > Q2 Similar in the weakness ...
>
> Please see our response to **W3**
>
> > Q3 The framework is ...
>
> We agree that understanding the contribution of each component is important. Our framework consists of three main components: (1) trajectory collection (including teacher/task selection and filtering), (2) diversity-preserving distillation (including KL with normalization), and (3) SSFT. In the paper, we have already analyzed the effects of teacher diversity, training task diversity, and SSFT interval. To further clarify the contribution of each module, we additionally conduct ablations on LES on BBOB by removing key components, including trajectory filtering, KL normalization, and SSFT. The results show that trajectory filtering and KL normalization account for the clearest degradation, while SSFT provides additional gains.
>
> | setting                                    | improvement over baseline |
> | ------------------------------------------ | ------------------------: |
> | MetaDistill                                |                    0.3602 |
> | w/ SSFT                                    |                    0.4031 |
> | w/o trajectory filtering (random segments) |                    0.1136 |
> | w/o KL norm                                |                    0.1196 |

---

> > ### Author Rebuttal · Reviewer_d5Dn · 2026-04-03
> >
> > Thank you for your clarification. I quite appreciate the basic idea of the paper of extracting the fundamental information trajectory from the teachers and utilise them for general distillation. And the experiment validation is rich.
> > However, some of the claims are a bit over especially when there are no theoretical foundations but only empirical observations. The claims like "Gaussian representation serves as an information bottleneck that filters out instance-specific or optimizer-specific details, and instead preserves the dominant search geometry (i.e., search center and principal directions)" and "promoting the acquisition of broader optimization principles" are a bit over claim in my own perspective.
> > How does the covariance and principal directions become effective information-carrying quantities? Why Gaussian? There lacks enough supporting arguments or references.
> > I would like to keep my current score but reduce my confidence since I am personally a little conflicted.

---

> > > ### Author Response · Authors · 2026-04-06
> > >
> > > We appreciate the reviewer for the follow-up comments. We hope to address your concerns in the following two aspects:
> > >
> > > **(1) How do mean and covariance become information carriers?** In black-box optimization (e.g., CMA-ES), modeling a population as a Gaussian distribution provides a compact yet sufficient representation of the search state. Specifically, the mean captures the current search center (i.e., where to exploit), while the covariance encodes the principal search directions (i.e., how to explore). Together, they form a minimal set of information required to drive optimization, as they directly determine both the location and geometry of search steps$^\star$. Importantly, this representation abstracts away instance-specific details of the landscape and instead preserves the dominant search geometry, which is largely shared across tasks. This makes it a natural candidate for transferable optimization strategies.
> > >
> > > **(2) Why Gaussian?** Following **(1)**, let's take POM as an example. If we discard the Gaussian distribution and turn to a more precise matching, that is, replacing the Gaussian KL divergence with the population-level mean square error in the distillation process, it will lead to a decrease in the generalization ability of POM.
> > >
> > > | loss                               | improvement over baseline |
> > > | ---------------------------------- | ------------------------: |
> > > | Gaussian KL divergence             |                    2.7607 |
> > > | Population-level mean square error |                    0.4653 |
> > >
> > > **Reference**
> > >
> > > $\star$ Hansen N. The CMA evolution strategy: A tutorial[J]. arXiv preprint arXiv:1604.00772, 2016.

---

### Decision · Program_Chairs · 2026-04-30

**Decision:**

Accept (regular)

**Comment:**

The reviewers and I unanimously support accepting the paper. It's a good training framework for training MetaBBO via distilling from BBO teachers. There were some initial concerns on comparing to baselines, discussing related L2O literature, and restricting to Gaussians that have been resolved after the discussion